# A Phase Transition between Positional and Semantic Learning in a Solvable Model of Dot-Product Attention

**Hugo Cui**
Statistical Physics of Computation laboratory
EPFL, Lausanne, Switzerland

**Freya Behrens**
Statistical Physics of Computation laboratory
EPFL, Lausanne, Switzerland

**Florent Krzakala**
Information, Learning and Physics laboratory
EPFL, Lausanne, Switzerland

**Lenka Zdeborová**
Statistical Physics of Computation laboratory
EPFL, Lausanne, Switzerland

## Abstract

Many empirical studies have provided evidence for the emergence of algorithmic mechanisms (abilities) in the learning of language models, that lead to qualitative improvements of the model capabilities. Yet, a theoretical characterization of how such mechanisms emerge remains elusive. In this paper, we take a step in this direction by providing a tight theoretical analysis of the emergence of semantic attention in a solvable model of dot-product attention. More precisely, we consider a non-linear self-attention layer with trainable tied and low-rank query and key matrices. In the asymptotic limit of high-dimensional data and a comparably large number of training samples we provide a tight closed-form characterization of the global minimum of the non-convex empirical loss landscape. We show that this minimum corresponds to either a positional attention mechanism (with tokens attending to each other based on their respective positions) or a semantic attention mechanism (with tokens attending to each other based on their meaning), and evidence an emergent phase transition from the former to the latter with increasing sample complexity. Finally, we compare the dot-product attention layer to a linear positional baseline, and show that it outperforms the latter using the semantic mechanism provided it has access to sufficient data.

## 1 Introduction

Recent years have seen an upheaval in our ability to learn and implement complex tasks from textual data. Instrumental in these advances is the use of self-attention layers [1], which provide an efficient method of extracting information from sentences – both the information encoded in the ordering (i.e. *positions*) of the words, and that encoded in the meaning (i.e. *semantics*) of the words. In theory, attention layers can learn to leverage both types of information, by having tokens attend to each other based on their respective positions (a mechanism called *positional attention* in [2]) and/or respective meanings (henceforth referred to as *semantic attention*). In this paper, we aim at a theoretical understanding of the emergence of these different mechanisms in attention layers, and the transitions therebetween.

Many empirical studies have provided evidence for the emergence of specific algorithmic mechanisms (abilities) in the learning of language models that lead to qualitative improvements of the model capabilities [3, 4, 5]. By reverse-engineering trained models into human-interpretable components [6, 7, 8] a growing body of work on mechanistic interpretability aims to empirically understand which precise algorithmic mechanisms a neural network learns. Such investigations have demonstrated that attention layers are able to implement a wide range of different algorithms, even for the same task, using both positional and semantic attributes of the inputs. We offer a particularly simple

38th Conference on Neural Information Processing Systems (NeurIPS 2024).

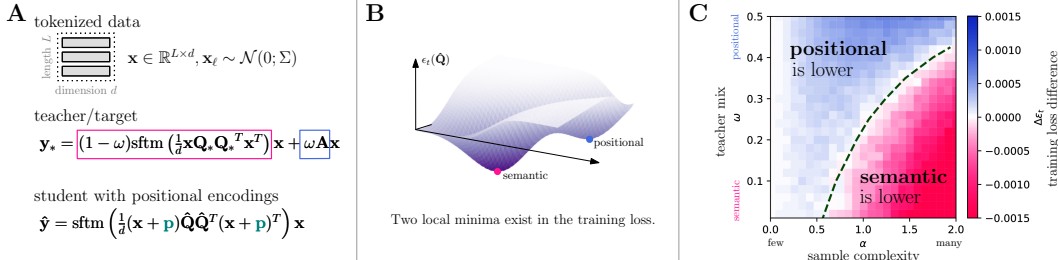

Figure 1: **A phase transition in a toy model of attention.** *(A)* We investigate a tied low-rank attention model in a teacher-student setting. The teacher mixes the $L$ individual tokens of dimension $d$ according to a semantic (as a function of the token's content $\mathbf{x}$) and a positional (as a function of the token's position) attention matrix. The student can only use positional encodings $\mathbf{p}$ to fit the positional properties of the teacher. *(B)* Schematic view of the loss landscape of the teacher, which contains both a positional and a semantic minimum. *(C)* We find that in the asymptotic high-dimensional limit and as a function of the sample complexity and the composition of the teacher, the global minimum switches, constituting a phase transition between positional and semantic learning.

illustration of this idea in Appendix A, where we show that for a sequence modelling task involving counting different algorithmic mechanisms co-exist, each corresponding to a distinct local minimum of the empirical loss. Out of these possibilities, the precise implementation that a model learns during training is jointly affected by its architecture [9], the training procedure itself [7, 4, 10] and the available data [11, 12, 13]. It remains an open question how to theoretically characterise the conditions under which a given behaviour emerges in the model leading to said qualitative improvements.

From a theoretical perspective, even the nature of this type of algorithmic emergence is unclear. It is not known whether it is simply a fast but smooth change in performance or whether the emergence is due to a sharp boundary between fundamentally different regimes of learning [14]. In our work, we take inspiration from physics, where a similar theoretical question about the nature of phase transitions was posed a century ago for models of interacting particles, such as the famous Ising model describing ferromagnetism [15, 16]. In the limit of infinitely many particles, it was shown that it is possible to theoretically deduce sharp discontinuities in some properties of interest (e.g. the magnetization of a magnet), delineating qualitatively very different regimes (e.g. magnetized or not). While mathematically, a large size limit needs to be considered to confirm the existence of sharp phase transitions, this asymptotic theory usually closely matches simulations, even for relatively moderate finite sizes.

In the theory of feed-forward fully connected neural networks, phase transitions in the network's generalisation ability as more training samples are available were studied as early as in [17, 18], and their existence was proven mathematically rigorously in [19]. In those works, the limit of many particles corresponds to taking the number of training samples and the dimensionality of the data to infinity at a fixed ratio. These theories rely on the property that macroscopic quantities of interest, such as the test error, become concentrated and deterministic in the high-dimensional limit. A dimension-free set of equations is then derived which predicts these deterministic quantities. Since then, a plethora of works along these lines both in statistical physics of phase transitions and in the theory of feed-forward neural networks have continued to study these phenomena, see e.g. [20, 21, 22, 23].

In this work, for the first time, we bring this type of study to the analysis of neural networks with attention layers. While several previous theoretical studies of the attention mechanism considered some type of high-dimensional limit [24, 25, 26, 2], none of them identified a phase transition between different types of mechanisms that are implemented by the attention layer. Simultaneously, the finite-dimensional real-world models that are the focus of works in mechanistic interpretability do not lend themselves to a tractable definition of a high-dimensional limit, which is necessary to identify a phase transition theoretically, as explained above. We aim to fill this gap by introducing and analysing a tractable model that permits such a sharp high-dimensional characterisation for attention layers (see Fig. 1).

In particular our contributions are:

- We describe a model with a single self-attention layer with tied, low-rank query and key matrices. On Gaussian input data and realizable labels, we show that *this model exhibits a phase transition in terms of sample complexity* between a semantic and a positional mechanism.

- As the main technical result, we analyse this model in the asymptotic limit where the embedding dimension $d$ of the tokens and the number $n$ of training samples grow proportionally. We provide a *tight closed-form characterization* of the test error and training loss achieved at the minima of the non-convex empirical loss. Using this high-dimensional characterization, we locate the positional-semantic phase transition, thus providing the first theoretical result about the *emergence of sharp phase transitions* in a model of dot-product attention.

- We contrast the performance of the dot-product attention layer with that of a linear model, which can only implement positional mechanisms, and show how the former outperforms the latter once it learns the semantic mechanism, *highlighting the advantage of the attention architecture* for this task, when there is a sufficient amount of training data.

In Section 2, we discuss further related work. Section 3 defines the general version of a solvable model of tied low-rank attention, and Section 4 provides a tight characterization of the global minimum of its empirical loss. Section 5 analyses a concrete instance of dot-product attention and demonstrates that in this case the global minimum corresponds to either a semantic or positional mechanism, depending on the training data and task, with a phase transition between them. We conclude with a discussion of the limitations of our analysis in Section 6.

## 2   Related work

**Theory of attention**   Attention models have been the object of sizeable theoretical scrutiny in recent years, with a growing body of work investigating various aspects such as their expressivity [25, 27, 28], inductive bias [29, 30, 31], training dynamics [2, 32, 33, 34], and in-context learning [26, 35, 36, 37, 38]. [39] and [25] analyze models with frozen non-trainable queries and keys, under the lens of signal propagation in such frozen models, or of their expressivity. [2] similarly studies the learning of the value matrix and positional encodings only, fixing keys and queries to identity, and shows how a transformer with a single attention head can learn spatial structure with a purely positional attention mechanism. The works of [29, 37] analyze the learning of a single layer of attention, with trainable queries and keys, assuming linear or ReLu activations – instead of the standard softmax activation. [33] provide convergence bounds for non-linear transformer models with a single attention layer, with trainable queries and keys. Because these studies are not tight, they do not allow to capture sharp changes in the behaviour of attention mechanisms such as phase transitions, and cannot for the same reason provide a theoretical description of the sudden emergence of new algorithmic mechanisms. A first tight description was provided in [24], in the context of learning a high-dimensional graphical model with a single layer of factored attention, leveraging its formal equivalence to a linear and convex learning problem. On the other hand, this model does not exhibit any emergent phenomenon. [38] analyze how induction head mechanisms can be learnt from gradient descent on the population loss, i.e. when an infinite amount of data is available. The present manuscript conducts a tight analysis of the non-convex learning of a non-linear attention model with trainable tied queries and keys from a finite train set, thereby allowing the description of sharp phase transitions in sample complexity in the behaviour and performance of the model.

**Positional encodings**   To combine the positional and semantic information in textual or general sequential data, a variety of models and input encodings have been explored. Many approaches are based on autoregressive models, e.g. recurrent architectures [40], where the positional information is provided implicitly by the order in which the input is processed. While some transformers can leverage implicit positional information through causal masks in training [41, 42, 43], in principle a dot product attention layer requires an explicit encoding of positional information as it views the input sequence in parallel, as a bag of words [1]. Several works experimentally explore different types of positional encodings with the goal of improving the downstream task performance [44, 45]. In this work, we provide a tractable model to quantify the generalization error of a single layer of attention in the presence of positional encodings.

**Theory of phase transitions in neural networks** In supervised learning with feed-forward fully connected neural networks, phase transitions in sample complexity were identified in settings where the data consists of random Gaussian samples, and the labels are generated by a target neural networks with random weights. For a single-layer perceptron and a variety of teacher weights distributions and activation functions, a discontinuity of the optimal test error as the number of samples increases was established in [18, 17, 19]. For a two-layer neural network, [21, 20] evidenced a specialization threshold in the sample complexity below which linear regression matches the optimal test accuracy, and above which a strictly better accuracy can be reached. To our awareness, phase transitions in neural networks with attention layers have not been studied theoretically yet.

## 3 Tied low-rank attention model

**Input data model** We consider a model of embedded sentences with uncorrelated (1-gram) words. More precisely, a sentence $\boldsymbol{x} \in \mathbb{R}^{L \times d}$, where $L$ is the sentence length and $d$ represents the embedding dimension, consists of $L$ tokens $\{\boldsymbol{x}_\ell\}_{1 \leq \ell \leq L}$ independently drawn from a Gaussian distribution $\boldsymbol{x}_\ell \sim \mathcal{N}(0, \boldsymbol{\Sigma}_\ell)$ with covariance $\boldsymbol{\Sigma}_\ell \in \mathbb{R}^{d \times d}$. In the following, we denote the probability distribution of $\boldsymbol{x}$ as $p_x$. Note that while this sentence model does not involve in itself statistical correlations between tokens, the task (target function) will entail interactions between different tokens. While more general data models involving inter-token correlations can also be readily analyzed such analyses come at the price of much more intricate analytical formulae. We thus choose for clarity to restrict the discussion to this simple instance, which already displays rich phenomenology, as will be explored in Section 5. We defer a discussion and an analytical treatment of the general case to Appendix C.

**Target function** The target function (teacher) is assumed to be of the form

$$y(x) = \mathtt{T}\left[\frac{1}{\sqrt{d}}\boldsymbol{x}\boldsymbol{Q}_\star\right]\boldsymbol{x}, \tag{1}$$

for a function $\mathtt{T} : \mathbb{R}^{L \times t} \to \mathbb{R}^{L \times L}$. The term $\mathtt{T}\left[1/\sqrt{d}\boldsymbol{x}\boldsymbol{Q}_\star\right] \in \mathbb{R}^{L \times L}$ in (1) should be interpreted as the target attention matrix, which mixes the tokens of the input $\boldsymbol{x}$. This attention matrix is parametrized by the target weights $\boldsymbol{Q}_\star \in \mathbb{R}^{d \times r_t}$.

**Tied attention** We consider the learning of the target (1) by a single attention layer

$$f_{\boldsymbol{Q}}(x) = \mathtt{S}\left[\frac{1}{\sqrt{d}}(\boldsymbol{x} + \boldsymbol{p})\boldsymbol{Q}\right](\boldsymbol{x} + \boldsymbol{p}). \tag{2}$$

In (2), $\boldsymbol{p} \in \mathbb{R}^{L \times d}$ is a *fixed* matrix, corresponding to positional encodings, and $\boldsymbol{Q} \in \mathbb{R}^{d \times r_s}$ is a trainable weight matrix. We denote subsequently $\boldsymbol{p}_\ell \in \mathbb{R}^d$ the $\ell-$th row of $\boldsymbol{p}$. Like the target (1), the parametric function (2) takes the form of a data-dependent attention matrix $\mathtt{S}\left[1/\sqrt{d}(\boldsymbol{x} + \boldsymbol{p})\boldsymbol{Q}\right] \in \mathbb{R}^{L \times L}$ mixing the tokens of the input $\boldsymbol{x}$. Note that, compared to the usual attention mechanism [1] employed in practice, (2) corresponds to setting the value weights to identity, and – since (2) is parametrized by a single matrix $\boldsymbol{Q}$– tying the key and query weights. While the assumption of tied weights is not strictly necessary, it makes for simpler and more interpretable analytical characterizations, and is thus considered in this work for clarity. We provide in Appendix C a full analysis of the untied architecture for completeness.

**Empirical risk minimization** We study the learning of the attention layer (2), when a training set $\mathcal{D} = \{\boldsymbol{x}^\mu, y(\boldsymbol{x}^\mu)\}_{\mu=1}^n$ with $n$ independently sampled sentences $\{\boldsymbol{x}^\mu\}_{\mu=1}^n$, and the associated labels $\{y(\boldsymbol{x}^\mu)\}_{\mu=1}^n$, is available. The target (1) can be learnt by carrying out an empirical risk minimization:

$$\hat{\boldsymbol{Q}} = \underset{\boldsymbol{Q} \in \mathbb{R}^{d \times r}}{\operatorname{argmin}}\left[\sum_{\mu=1}^n \frac{1}{2d}\|y(\boldsymbol{x}^\mu) - f_{\boldsymbol{Q}}(\boldsymbol{x}^\mu)\|^2 + \frac{\lambda}{2}\|\boldsymbol{Q}\|^2\right]. \tag{3}$$

The performance of the resulting trained model $f_{\hat{\boldsymbol{Q}}}$ is measured at test time by the mean squared error (MSE)

$$\epsilon_g \equiv \frac{1}{dL}\mathbb{E}_{\boldsymbol{x} \sim p_x}\left\|y(\boldsymbol{x}) - f_{\hat{\boldsymbol{Q}}}(\boldsymbol{x})\right\|^2. \tag{4}$$

# 4   Closed-form characterization of the training loss

**High-dimensional limit**   We analyze the learning problem (3) in the limit where the embedding dimension $d$ and the number of training samples $n$ jointly tend to infinity, while their ratio $\alpha = n/d$ (henceforth referred to as the sample complexity) stays of order $\Theta_d(1)$. We further assume the sentence length $L$, the ranks $r_s, r_t$ of the weights $Q, Q_\star$, and the norm of the positional embeddings $\|p\|$, to be $\Theta_d(1)$. This limit has been considered in a stream of previous works (e.g. [46, 47, 48]) and allows to derive closed-form characterization of the ERM problem (3), which we present in the next section. It also exhibits a particularly rich learning phenomenology which we further explore in Section (5). Finally, let us comment briefly on the assumption that $r_s = \Theta_d(1)$, which in words implies that the weight matrix $Q$ is *low-rank*. While primarily motivated by technical limitations here, it is worth noting that low-rank weights are also considered in machine learning practice, in the context of model compression [49] or fine-tuning [50].

The **main technical result** of the present work is a closed-formed characterization of the test MSE (4) and training loss (3) achieved in the high-dimensional limit when training the model (2) via the empirical risk minimization of (3).

**Assumption 4.1.** *The covariances $\{\Sigma_\ell\}_{\ell=1}^L$ admit a common set of eigenvectors $\{e_i\}_{i=1}^d$. We further note $\{\lambda_i^\ell\}_{i=1}^d$ the eigenvalues of $\Sigma_\ell$. The eigenvalues $\{\lambda_i^\ell\}_{i=1}^d$ and the projection of the positional embedding $\{p_\ell\}_{\ell=1}^L$ and the teacher columns $\{Q_j^\star\}_{j=1}^{r_t}$ on the eigenvectors $\{e_i^\top p_\ell\}_{i,\ell}$, $\{e_i^\top Q_j^\star\}_{i,j}$ are assumed to admit a well-defined joint distribution $\nu$ as $d \to \infty$ – namely, for $\gamma = (\gamma_1, ..., \gamma_L) \in \mathbb{R}^L, \pi = (\pi_1, ..., \pi_{r_t}) \in \mathbb{R}^{r_t}$ and $\tau = (\tau_1, ..., \tau_L) \in \mathbb{R}^L$:*

$$\frac{1}{d} \sum_{i=1}^d \prod_{\ell=1}^L \delta\left(\lambda_i^\ell - \gamma_\ell\right) \delta\left(\sqrt{d}e_i^\top p_\ell - \tau_\ell\right) \prod_{j=1}^{r_t} \delta\left(e_i^\top Q_j^\star - \pi_j\right) \xrightarrow{d \to \infty} \nu\left(\gamma, \tau, \pi\right). \tag{5}$$

In words, Assumption 4.1 guarantees that all parameters of the problem admit well-defined limits in the considered asymptotic limit, with the further assumption that the covariances $\{\Sigma_\ell\}_{\ell=1}^L$ of the different tokens can be jointly diagonalized. We are now in a position to state the main technical result of the present work.

**Result 4.2.** *Under Assumption 4.1, in the limit $n, d \to \infty$, $\|p\|, n/d, L, r_s, r_t = \Theta_d(1)$, the summary statistics*

$$\rho_\ell \equiv \frac{Q_\star^\top \Sigma_\ell Q_\star}{d} \in \mathbb{R}^{r_t \times r_t}, \qquad\qquad q_\ell \equiv \frac{\hat{Q}^\top \Sigma_\ell \hat{Q}}{d} \in \mathbb{R}^{r_s \times r_s},$$

$$m_\ell \equiv \frac{\hat{Q}^\top p_\ell}{d} \in \mathbb{R}^{r_s}, \qquad\qquad \theta_\ell \equiv \frac{\hat{Q}^\top \Sigma_\ell Q_\star}{d} \in \mathbb{R}^{r_s \times r_t} \tag{6}$$

*concentrate in probability, and are solutions of the set of finite-dimensional self-consistent equations*

$$\begin{cases} q_\ell = \int d\nu(\gamma, \tau, \pi)\gamma_\ell \left(\lambda \mathbb{I}_{r_s} + \sum_{\kappa=1}^L \gamma_\kappa \hat{V}_\kappa\right)^{-1} \left(\sum_{\kappa=1}^L \gamma_\kappa \hat{q}_\kappa + \left(\sum_{\kappa=1}^L \hat{m}_\kappa \tau_\kappa + \gamma_\kappa \hat{\theta}_\kappa \cdot \pi\right)^{\otimes 2}\right) \left(\lambda \mathbb{I}_{r_s} + \sum_{\kappa=1}^L \gamma_\kappa \hat{V}_\kappa\right)^{-1} \\ V_\ell = \int d\nu(\gamma, \tau, \pi)\gamma_\ell \left(\lambda \mathbb{I}_{r_s} + \sum_{\kappa=1}^L \gamma_\kappa \hat{V}_\kappa\right)^{-1} \\ m_\ell = \int d\nu(\gamma, \tau, \pi)\tau_\ell \left(\lambda \mathbb{I}_{r_s} + \sum_{\kappa=1}^L \gamma_\kappa \hat{V}_\kappa\right)^{-1} \left(\sum_{\kappa=1}^L \hat{m}_\kappa \tau_\kappa + \gamma_\kappa \hat{\theta}_\kappa \cdot \pi\right) \\ \theta_\ell = \int d\nu(\gamma, \tau, \pi)\gamma_\ell \left(\lambda \mathbb{I}_{r_s} + \sum_{\kappa=1}^L \gamma_\kappa \hat{V}_\kappa\right)^{-1} \left(\sum_{\kappa=1}^L \hat{m}_\kappa \tau_\kappa + \gamma_\kappa \hat{\theta}_\kappa \cdot \pi\right) \pi^\top. \end{cases} \tag{7}$$

$$\begin{cases} \hat{q}_\ell = \alpha \mathbb{E}_{\Xi, U} V_\ell^{-1} \left(\text{prox}(\Xi, U)_\ell - q_\ell^{\frac{1}{2}} \xi_\ell - m_\ell\right)^{\otimes 2} V_\ell^{-1} \\ \hat{V}_\ell = \hat{\theta}_\ell \theta_\ell^\top q_\ell^{-1} - \alpha \mathbb{E}_{\Xi, U} V_\ell^{-1} \left(\text{prox}(\Xi, U)_\ell - q_\ell^{\frac{1}{2}} \xi_\ell - m_\ell\right) \xi_\ell^\top q_\ell^{-\frac{1}{2}} \\ \hat{m}_\ell = \alpha \mathbb{E}_{\Xi, U} V_\ell^{-1} \left(\text{prox}(\Xi, U)_\ell - q_\ell^{\frac{1}{2}} \xi_\ell - m_\ell\right) \\ \hat{\theta}_\ell = \alpha \mathbb{E}_{\Xi, U} V_\ell^{-1} \left(\text{prox}(\Xi, U)_\ell - q_\ell^{\frac{1}{2}} \xi_\ell - m_\ell\right) \left(u_\ell - \xi_\ell^\top q_\ell^{-1/2} \theta_\ell\right)^\top \left(\rho_\ell - \theta_\ell^\top q_\ell^{-1} \theta_\ell\right)^{-1} \end{cases} \tag{8}$$

In (7), $U = \{u_\ell\}_{\ell=1}^L$ and $\Xi = \{\xi_\ell\}_{\ell=1}^L$, with $u_\ell \sim \mathcal{N}(\xi_\ell^\top q_\ell^{-1/2}\theta_\ell, \rho_\ell - \theta_\ell^\top q_\ell^{-1}\theta_\ell)$ and $\xi_\ell \sim \mathcal{N}(0, \mathbb{I}_{r_s})$, and $\cdot^{\otimes 2}$ denotes the outer product of a vector with itself. Finally, the resolvents $\{\mathrm{prox}(\Xi, U)_\ell\}_{\ell=1}^L$ are defined as the minimizers of the Moreau envelope

$$\mathcal{M}(\Xi, U) = \inf_{z_1,\ldots,z_L} \left\{ \sum_{\ell=1}^L \mathrm{Tr}\left[ V_\ell^{-1} \left( x_\ell - q_\ell^{1/2}\xi_\ell - m_\ell \right)^{\otimes 2} \right] + \mathrm{Tr}\left[ \mathtt{S}(Z)\rho_\Sigma \mathtt{S}(Z)^\top \right] - 2\mathrm{Tr}\left[ \mathtt{T}(U)\rho_\Sigma \mathtt{S}(Z)^\top \right] \right\}.$$

We noted $Z \in \mathbb{R}^{L\times r_s}$ (resp. $U \in \mathbb{R}^{L\times r_t}$) the matrix whose rows are $z_\ell$ (resp. $u_\ell$) and:

$$\rho_\Sigma \equiv \mathrm{diag}\left[ \left( \int d\nu(\gamma,\tau,\pi)\gamma_\ell \right)_{\ell=1}^L \right] \in \mathbb{R}^{L\times L}. \tag{9}$$

In the same limit, the test error (4) converges in probability to

$$\epsilon_g = \frac{1}{L}\mathbb{E}_h \, \mathrm{Tr}\left[ \mathtt{S}[h]\rho_\Sigma \mathtt{S}[h]^\top \right] + \frac{1}{L}\mathbb{E}_{h^\star} \, \mathrm{Tr}\left[ \mathtt{T}[h^\star]\rho_\Sigma \mathtt{T}[h^\star]^\top \right] - 2\frac{1}{L}\mathbb{E}_{h,h^\star} \, \mathrm{Tr}\left[ \mathtt{S}[h]\rho_\Sigma \mathtt{T}[h^\star]^\top \right]. \tag{10}$$

where the average bears on $h \in \mathbb{R}^{L\times r_s}, h^\star \in \mathbb{R}^{L\times r_t}$ with independent rows with statistics

$$(h_\ell, h_\ell^\star) \sim \mathcal{N}\left[ \begin{pmatrix} m_\ell \\ 0 \end{pmatrix}, \left( \begin{array}{c|c} q_\ell & \theta_\ell \\ \hline \theta_\ell^\top & \rho_\ell \end{array} \right) \right] \tag{11}$$

Finally, the training loss $\epsilon_t$ converges in probability to

$$\epsilon_t = \alpha\mathbb{E}_{Y,\Xi}\mathcal{M} - \frac{1}{2}\sum_{\ell=1}^L \mathrm{Tr}[\hat{q}_\ell V_\ell] + \frac{\lambda}{2}\int d\nu(\gamma,\tau)\mathrm{Tr}\left[ \left( \lambda + \sum_{\ell=1}^L \gamma_\ell \hat{v}_\ell \right)^{-2}\left( \sum_{\ell=1}^L \gamma_\ell \hat{q}_\ell + \left( \sum_{\ell=1}^L \tau_\ell \hat{m}_\ell + \hat{\theta}_\ell \cdot \pi \right)^{\otimes 2} \right) \right]. \tag{12}$$

Result 4.2 provides a tight asymptotic characterization of the test error $\epsilon_g$ (10) and the training loss $\epsilon_t$ (12), as a function of a finite set finite-dimensional summary statistics $\{q_\ell, V_\ell, m_\ell, \theta_\ell\}_{\ell=1}^L$, thereby providing a finite-dimensional description of the high-dimensional learning problem (3). These summary statistics are further characterized in closed form by the set of equations (7), in terms of the solution of a low-dimensional minimization problem (91). Intuitively, this low-dimensional problem may be viewed as a form of an effective loss averaged over the finite training set. In practice, the solution of the self-consistent equations (7) can be found by numerically iterating the equations until convergence. The resulting summary statistics can then be used to evaluate the expressions (10) and (12) to reach the asymptotic limits of the test error and train loss. In Section 5 we evaluate exactly these functions to understand the different minima in the models empirical loss landscape. Similar sharp asymptotic characterizations have been derived in the literature for other neural network architectures trained with ERM, in particular generalized linear models (see e.g. [51, 52, 48]) and auto-encoders [47].

The derivation of Result 4.2 is provided in Appendix B, and is exploiting a mapping of the model (2) to a (variant of) a Generalized Linear Model (GLM) [53, 54]. The summary statistics characterized by the equations (7) (often called state evolution [55] in this context) asymptotically describe the fixed points of a Generalized Approximate Message Passing (GAMP) algorithm [56], which we state in Appendix B. The fixed points of GAMP in turn correspond to critical (zero-gradient) points of the non-convex empirical loss landscape (3). Therefore, while Result (4.2) is stated as a characterization of the global minimum of (3), which is the main concern of the present work, solutions of (7) also describe local minima and saddles.

This strategy has been used in many recent work to study asymptotics of a large number of high-dimensional problems, see e.g. [57, 58, 59, 60, 48]. We note, however, that we importantly assume the point-wise convergence of GAMP. While we believe that this point can be rigorously justified, it would require a considerable amount of work —in particular, the usual rigorous tools used in recent works fall short because of the non-convexity of the loss— and we leave this point for future studies (see the discussion in Appendix C, where we also provide an alternative derivation using the replica method from statistical physics [61]). Finally, we mention that while Result 4.2 is presented for an $\ell_2$ regularization of the empirical loss (3) for clarity, similar results can be reached for generic convex regularizers, and are presented in Appendix C. In the following section, we explore the phenomenology uncovered from the study of the equations (7) of Result 4.2, for the special case of dot-product attention.

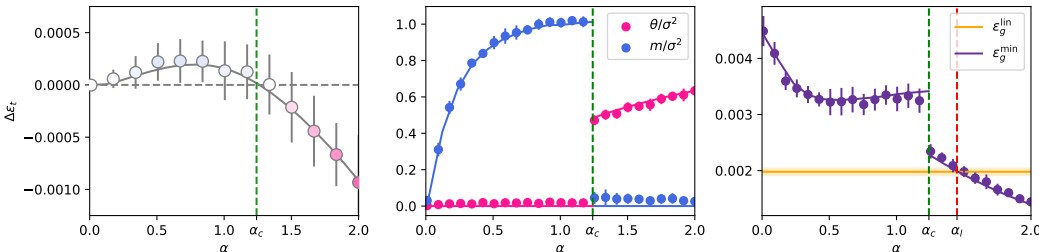

Figure 2: **Mixed positional/semantic teacher for** $\omega = 0.3$. Setting is $r_s = r_t = 1, L = 2, A = ((0.6, 0.4), (0.4, 0.6)), \Sigma_1 = \Sigma_2 = 0.25\mathbb{I}_d, p_1 = \mathbf{1}_d = -p_2$ and $Q_\star \sim \mathcal{N}(0, \mathbb{I}_d)$. **(left)** Solid lines: difference in training loss $\Delta\epsilon_t$ between the semantic and positional solutions of (7) in Result 4.2. Markers: difference in training loss at convergence achieved by training the model (2) using gradient descent initialized resp. at $Q_\star$ and at $p_1$. Marker color as in Fig. 3. **(center)** overlap $\theta$ between the learnt weights $\hat{Q}$ and the target weights $Q_\star$ overlap $m$ between the learnt weights $\hat{Q}$ and the positional embedding $p_1$. Solid lines represent the theoretical characterization of these two summary statistics provided by Result 4.2. Only the solution of (7) corresponding to the lowest found training loss is represented (i.e. the positional solution for $\alpha < \alpha_c$ and the semantic otherwise). Markers represent experimental measures of these quantities, for gradient descent at convergence. Gradient descent was initialized at $p_1$ for $\alpha < \alpha_c$ and at $Q_\star$ for $\alpha > \alpha_c$. **(right)** We show the MSE achieved by the dense linear as $\epsilon_g^{min}$ (Result 4.2), and MSE achieved by the dense linear baseline $\epsilon_g^{lin}$ (15) (Result 5.1). Markers indicate the MSE experimentally reached by the model (2) trained using gradient descent, initialized previously for the overlaps. All experiments were performed in $d = 1,000$ with the Pytorch implementation of full-batch gradient descent, for $T = 5,000$ epochs and learning rate $\eta = 0.15$. All points are averaged over $24$ instances of the problem each.

## 5 Positional-to-semantic phase transition

**Rank one dot-product attention** In the following, we turn to a special case of tied low-rank attention (2) – namely a dot-product attention layer, which is the example from Fig. 1:

$$\mathbf{S}\left[\frac{1}{\sqrt{d}}(\boldsymbol{x} + \boldsymbol{p})\boldsymbol{Q}\right] = \text{softmax}\left(\frac{1}{d}(\boldsymbol{x} + \boldsymbol{p})\boldsymbol{Q}\boldsymbol{Q}^\top(\boldsymbol{x} + \boldsymbol{p})^\top\right). \quad (13)$$

As in (2), we allow for positional encodings $\boldsymbol{p}$ in the dot-product attention parametrization (13). We further consider a specific case of target attention matrix (1) of the form

$$\mathbf{T}\left[\frac{1}{\sqrt{d}}\boldsymbol{x}\boldsymbol{Q}_\star\right] = (1 - \omega)\text{softmax}\left(\frac{1}{d}\boldsymbol{x}\boldsymbol{Q}_\star\boldsymbol{Q}_\star^\top\boldsymbol{x}^\top\right) + \omega A. \quad (14)$$

with $A \in \mathbb{R}^{L \times L}$ a fixed matrix. In (14), the parameter $\omega \in [0, 1]$ tunes the relative strength of the dot-product term and the fixed matrix term, and interpolates between a fully positional and a fully semantic task:

- For $\omega = 0$, the target reduces to the first dot-product term, and is purely semantic, in that the $i, j-$th element of the score matrix $\text{softmax}(1/d\boldsymbol{x}\boldsymbol{Q}_\star\boldsymbol{Q}_\star^\top\boldsymbol{x}^\top)$ only depends on the tokens $\boldsymbol{x}_i, \boldsymbol{x}_j$ and not explicitly on their respective placements $i, j$ inside the sentence. To learn satisfyingly the target, the learning model thus has to learn a *semantic* attention matrix.

- For $\omega = 1$, the target reduces to the second fixed term $A$ in (14). The attention matrix $A$ associated thereto is purely positional, in the sense that $A_{ij}$ is a function of $i, j$ but not of $\boldsymbol{x}_i, \boldsymbol{x}_j$. To complete the learning task, a *positional* mechanism then needs to be learnt.

The parameter $\omega$ thus allows to tune the amount of semantic/positional content in the target (14), and thus the extent to which the task requires the model to implement semantic attention (small $\omega$s) or rather positional attention (large $\omega$s). In the following, for definiteness, we further assume $r_s = r_t = 1$ and set $Q_\star$ to be a fixed random Gaussian vector drawn from $\mathcal{N}(0, \mathbb{I}_d)$, and choose the positional encodings $p_1 = -p_2 = \mathbf{1}_d$. Finally, for simplicity, we consider sentences with two tokens $L = 2$ and isotropic token covariances $\Sigma_1 = \Sigma_2 = \sigma^2\mathbb{I}_d$.

**Semantic and positional mechanisms** The summary statistics $\theta_\ell, m_\ell$ describing the global minimizer of the empirical loss minimization (3) of the dot-product attention (13) on the target

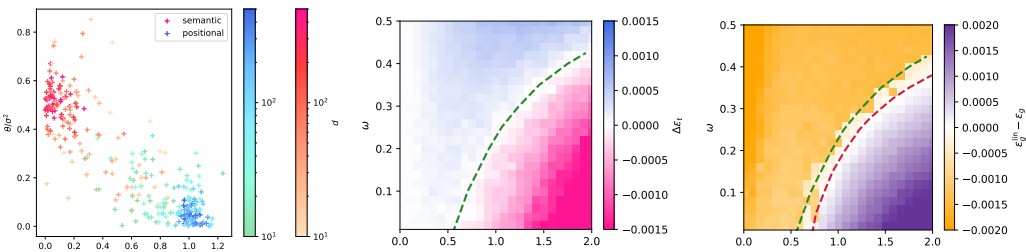

Figure 3: **Phase transition between semantic and positional training loss.** Setting and experiments were performed identical to Fig. 2. **(left)** Scaling $d$ and $n$ jointly for $\alpha = 1.5$ concentrates for $\theta$ and $m$, in different locations for the positional and semantic local minima each. We show 30 runs for each $d \in [10, 15, 23, 36, 56, 87, 135, 209, 323, 500]$. **(center)** The color map represents the difference in training loss at convergence when training the model (2) using the `Pytorch` implementation of full-batch gradient descent, respectively from an initialization at $\mathbf{p}_1$ or at $\boldsymbol{Q}_\star$. The green dashed line represents the theoretical prediction for the threshold $\alpha_c(\omega)$ above which the semantic solution of (7) in Result 4.2 has lower loss than the positional solution. **(right)** The color map represents the difference in test MSE at convergence when training the attention model (13) using the `Pytorch` implementation of full-batch gradient descent initialized at $\boldsymbol{Q}_\star$, and the dense linear baseline (15). The red dashed lines indicate the theoretical prediction –following from Result 4.2 and Result 15– for the threshold sample complexity $\alpha_l(\omega)$ above which the dot-product attention (2) outperforms the baseline (15).

(14) are captured alongside the corresponding test error (4) and training loss (3), by Result 4.2. The solution of the system of equations (7) is not unique, and different stable fixed points describe different corresponding critical points of the non-convex empirical loss landscape (3). In practice, we notably find two solutions of (7), corresponding to two minima associated with different mechanisms implemented by the dot-product attention (13) when approximating the target (14):

*–Positional solution* One solution of (7) correspond to vanishing overlap $\theta = 0$ between the trained weights $\hat{\boldsymbol{Q}}$ and the semantic target weights $\boldsymbol{Q}_\star$, and non-zero $m > 0$ between the trained weights $\hat{\boldsymbol{Q}}$ and the positional embedding $\boldsymbol{p}_1 = -\boldsymbol{p}_2$. Consequently, the argument of the dot-product attention $\hat{\boldsymbol{Q}}(\boldsymbol{x} + \boldsymbol{p})$ has a sizeable token-independent –thus positional– contribution $\hat{\boldsymbol{Q}}\boldsymbol{p}$, alongside a token-dependent semantic part $\hat{\boldsymbol{Q}}\boldsymbol{x}$. Because of the positional terms, the resulting learnt attention attention matrix $\mathrm{softmax}(1/d(\boldsymbol{x} + \boldsymbol{p})\hat{\boldsymbol{Q}}\hat{\boldsymbol{Q}}^\top(\boldsymbol{x} + \boldsymbol{p})^\top)$ implements a partly positional mechanism.
*–Semantic solution* Another solution of the system of equations (7) is associated with a vanishing overlap $m = 0$ between the learnt weights $\hat{\boldsymbol{Q}}$ and the positional embeddings, and a finite overlap $\theta > 0$ with the target weights $\boldsymbol{Q}_\star$. Therefore the resulting learnt attention matrix $\mathrm{softmax}(1/d(\boldsymbol{x} + \boldsymbol{p})\hat{\boldsymbol{Q}}\hat{\boldsymbol{Q}}^\top(\boldsymbol{x} + \boldsymbol{p})^\top) \approx \mathrm{softmax}(1/d\boldsymbol{x}\hat{\boldsymbol{Q}}\hat{\boldsymbol{Q}}^\top\boldsymbol{x}^\top)$ is largely semantic.

While the system of self-consistent equations (7) may admit other solutions, we did not find solutions with lower training loss than the two aforedescribed fixed points. Which of these solution corresponds to the global minimum – and thus the solution of the optimization (3)– depends on the sample complexity $\alpha$ and the positional/semantic parameter $\omega$ (14), as we describe in the following subsection.

**Positional-to-semantic phase transition** For a fixed parameter $\omega$ in (14), an analysis of equations (7), further detailed in Appendix C, reveals that for a sizeable range of $\omega$, in the probed setups, there exists a threshold $\alpha_c$ for the sample complexity so that

- For $\alpha < \alpha_c$, the global minimum of (3) corresponds to a positional mechanism, and is described by the positional solution of (7) of Result 4.2 with $\theta = 0, m > 0$.
- For $\alpha > \alpha_c$, the global minimum of (3) corresponds to a semantic mechanism, and is described by the semantic solution of (7) of Result 4.2 with $\theta > 0, m = 0$.

The dot-product attention thus displays *a phase transition in sample complexity from a positional mechanism to a semantic mechanism*, implementing the simpler positional mechanism when having access to small amounts of data, and only learning the semantic content of the target (14) when presented sufficient data. The critical sample complexity $\alpha_c$ generically grows with the positionality $\omega$ of the target function (14), as the semantic content – i.e. the first term of (14)– is less apparent for larger $\omega$, and thus requires larger amounts of data to be identified and approximated by the dot-product attention (13). An example for $\omega = 0.3$ is given in Fig. 2. In Fig. 3 (center) the difference

in training loss $\Delta \epsilon_t$ between the positional and semantic solutions of (7) is represented, alongside the difference in training loss at convergence experimentally reached by gradient descent. For small (resp. large) sample complexity $\alpha < \alpha_c$ (resp. $\alpha > \alpha_c$), the training loss of the positional (resp. semantic) minimum is lower, and thus corresponds to the global minimum.

Experimentally, the positional minimum can be reached for $\alpha < \alpha_c$ via gradient descent by initializing the weights $\boldsymbol{Q}$ of the attention (13) close to the positional embedding $\boldsymbol{p}_1$. By the same means, the semantic minimum can be reached from an initialization at the teacher weights $\boldsymbol{Q}_\star$ (14). Henceforth, we refer with a slight abuse to the minimum experimentally reached from a positional (resp. semantic) initialization as the positional (resp. semantic) minimum, even when it is not global. Note that importantly the semantic initialization is informed in nature, in that it necessitates the knowledge of the target parameters $\boldsymbol{Q}_\star$. Note that even though the minima we characterize analytically are fixed points of gradient descent, a precise analysis of the *dynamics* of gradient descent from an agnostic (random) initialization, and ascertaining whether the optimizer reaches the global minimum, is an interesting question but falls out of the scope of the present manuscript – which is an analysis of the loss landscape. We however conduct numerical experiments from a random initialization of $\boldsymbol{Q}$ in Appendix E.4, and show that the dynamics may reach either of the local minima, or get stuck in a different one.

In Fig. 2, we compare our analytical characterizations for different metrics at the global mimimum – the summary statistics $\theta, m$ (middle), and the test MSE (right)–, with the corresponding experimental estimates, obtained by optimizing (3) with the `Pytorch` implementation of gradient descent, from a positional (resp. semantic) initialization for $\alpha < \alpha_c$ (resp. $\alpha > \alpha_c$), displaying overall good agreement. In Fig. 3 (left) and Appendix E.1 we further verify that in the scaling limit of our analysis, namely $n, d \to \infty$ for $\alpha = O(1)$, the agreement improves with growing $n, d$.

The dot-product attention (13) thus implements a semantic mechanism when learning from sufficient amounts of data. The learning of the semantic mechanism by the dot-product attention at sample complexities $\alpha > \alpha_c$ corresponds to a noticeable drop in the generalization MSE as can be observed in Fig. 2, right. But just how essential is the learning of semantic mechanism in the ability of the dot-product attention to generalize well? We explore this question in the following subsection, by comparing the dot-product attention (13) to a purely positional attention model.

**Purely positional baseline** In this subsection, for the same target (14), we contrast the dot-product attention model (13), analyzed in the previous subsections, to the baseline given by a linear layer

$$f_W(\boldsymbol{x}) = W \cdot \boldsymbol{x}, \tag{15}$$

with a trainable weight matrix $W \in \mathbb{R}^{L \times L}$. As for the dot-product attention (13), we consider the case where the weights $\hat{W}$ are learnt by minimizing the empirical risk

$$\hat{W} = \underset{W \in \mathbb{R}^{L \times L}}{\operatorname{argmin}} \sum_{\mu=1}^{n} \|y(\boldsymbol{x}^\mu) - f_W(\boldsymbol{x}^\mu)\|^2. \tag{16}$$

The model (15) is a natural counterpart to the dot-product architecture (13). In (15), the attention matrix is parametrized by a single fully-trainable matrix $W$, instead of being parametrized as a dot-product attention as in (13). A seminal difference in the two parametrizations is that while the elements of $\operatorname{softmax}(1/d\boldsymbol{x}\boldsymbol{Q}\boldsymbol{Q}^\top \boldsymbol{x}^\top)$ can depend on the input tokens $\boldsymbol{x}$, and therefore express semantic information, the elements $W_{ij}$ of $W$ can only depend on the positions $i, j$. The model (15) can thus only implement *positional mechanisms*, while the dot-product attention (13) can implement both linear and semantic mechanisms, as discussed above. Finally, observe that the model (15) is closely related to the one analyzed by [24] in another asymptotic limit. The following result characterizes the test error achieved by the purely positional model (15):

**Result 5.1.** *In the same asymptotic limit as Result* (4.2)*, the learnt weights $\hat{W}$ trained by minimizing the empirical risk* (16) *coincide with the minimizer of the population risk, and thus admit the compact expression*

$$\hat{W} = \mathbb{E}_{\boldsymbol{x}} \mathtt{T} \left[ \frac{1}{\sqrt{d}} \boldsymbol{x} \boldsymbol{Q}_\star \right] = \mathbb{E}_h \mathtt{T}[h] \tag{17}$$

*where the average bears over a finite-dimensional matrix $h \in \mathbb{R}^{L \times t}$ with independent rows $h_\ell$ with statistics $h_\ell \sim \mathcal{N}(0, \rho_\ell)$, where $\rho_\ell$ was defined in* (6) *in Result* (4.2)*. We remind that $\mathtt{T} \left[ 1/\sqrt{d}\boldsymbol{x}\boldsymbol{Q}_\star \right]$*

*corresponds to the target score matrix* (1). *Finally, the test MSE* $1/dL\mathbb{E}_{\boldsymbol{x}}\|y(\boldsymbol{x}) - f_{\hat{W}}(\boldsymbol{x})\|^2$ *achieved by the trained dense linear model* $f_{\hat{W}}$ (15) *admits the asymptotic characterization*

$$\epsilon_g^{\text{lin}} = \frac{1}{L} \operatorname{Tr}\left[\hat{W}\rho_\Sigma\hat{W}^\top\right] + \frac{1}{L}\mathbb{E}_h \operatorname{Tr}\left[\texttt{T}[h]\rho_\Sigma\texttt{T}[h]^\top\right] - \frac{2}{L}\mathbb{E}_h \operatorname{Tr}\left[\hat{W}\rho_\Sigma\texttt{T}[h]^\top\right]. \tag{18}$$

The MSE achieved by the baseline (15) when learning the target (14) is plotted in Fig. 2 (right) as the orange solid line, alongside the MSE achieved by the dot-product attention (13) discussed in previous subsections. Remarkably, in the setup of Fig. 2, in the positional regime $\alpha < \alpha_c$ when the dot-product attention relies on a positional mechanism $\theta = 0, m > 0$ to approximate the target, the dot-product attention (13) is outperformed by the purely positional attention (15) $\epsilon_g > \epsilon_g^{\text{lin}}$. In contrast, in the semantic regime $\alpha > \alpha_c$ where the dot-product attention learns the semantic mechanism, there exists a sample complexity $\alpha_l \geq \alpha_c$ above which $\epsilon_g < \epsilon_g^{\text{lin}}$, i.e. the dot-product attention (13) outperforms the dense linear baseline (15). This threshold value $\alpha_l$ is plotted for various positionality strengths $\omega$ in Fig. 3, alongside the positional-to-semantic threshold $\alpha_c$. Interestingly, we observe $\alpha_l \geq \alpha_c$ in all probed settings, temptingly suggesting the natural interpretation that the dot-product attention needs to learn the semantic mechanism first (at $\alpha = \alpha_c$) in order to then be able to outperform the best positional approximation $f_{\hat{W}}$ (at $\alpha = \alpha_l$). This highlights the importance of the semantic mechanism, enabled by the dot-product parametrization (13), in learning targets with semantic content such as (14).

## 6   Limitations

Compared to the original transformer [1] we consider a *simplified model*: the query and key matrices in our model are sharing weights and are of a low rank, as a value matrix we use the identity, and we employ only one head and a single layer. Further, our data model is limited to Gaussian data with sentences of 1-grams. Concerning the analysis, our characterization holds only in the *high-dimensional limit*, but we show that even in the finite case experimental values lie close to the theoretical prediction. Since the analysis *only concerns the minima of the models loss landscape* the implications for the dynamics of learning algorithms, e.g. gradient decent, are limited. This shows in our numerical experiments where we need to initialize GD close to the minima in order to arrive at them. It is yet unclear if there are scaling limits of learning algorithms which would reliably find the lower or a specific one of the minima from a random initialization.

## Conclusion

We explored the interplay between positional and semantic attention, through the prism of tied low-rank self-attention in high dimensions. In a theoretically controlled setting, we characterized the global optimum of the empirical loss, when learning a target attention layer. This global optimum was found to correspond to either a positional or a semantic mechanism, with a phase transition between the two mechanisms occurring as the sample complexity increases. We believe the present asymptotic analysis of the inner workings of attention mechanisms opens up exciting research directions. Considering alternative attention architectures (including a readout network after the attention layer, or considering cross-attention) or training procedures (such as masked language modelling, or training with causal masks), are some possible extensions which will hopefully pave the way towards a satisfactory theoretical comprehension of attention mechanisms. Finally, elucidating under which conditions either minimum can be reached by a given optimizer from a random initialization constitutes an important future research avenue.

## Acknowledgements

We thank Luca Biggio, Federica Gerace, and Matteo Vilucchio for insightful discussions. We acknowledge funding from the Swiss National Science Foundation grant SNFS OperaGOST (grant number 200390), and SMArtNet (grant number 212049).

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

# Appendix: Table of Contents

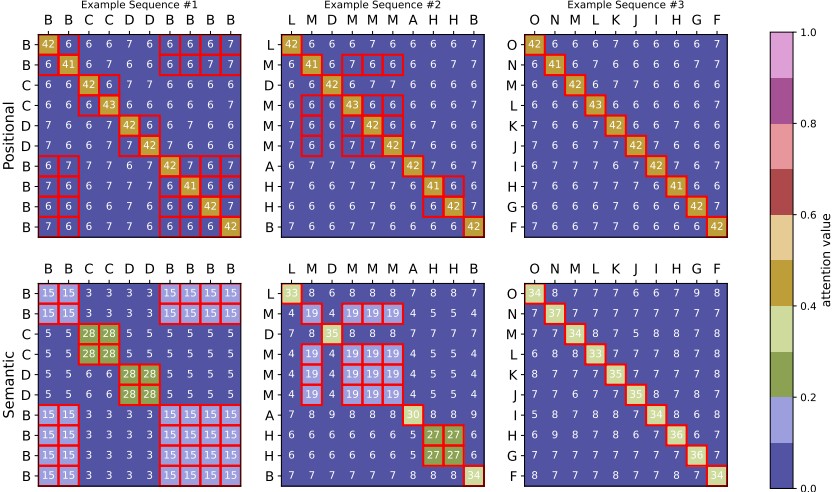

Figure 4: **Several solutions exist for the histogram task.** Elements of attention matrices for the histogram task for local minima in the empirical loss landscape. We generated a dataset of sequences by sampling each token of the sequence i.i.d. from the uniform distribution over all tokens. The target of a a given input sequence $\mathbf{x} = [A, D, D, C]$ is the number of occurence of each token in the complete sequence, i.e. $\mathbf{y} = [1, 2, 2, 1]$. Models were trained with their respective frozen initialization using $n = 35,000$ samples and the Adam optimizer. *Top Row:* The attention matrix of the positional solution is largely independent of the specific input sequence. *Bottom Row:* The attention matrices from the semantic solution vary based on the input token. Red squares highlight the elements of $A_{ij}$ where $x_i = x_j$.

# A   The Histogram Task: An algorithmic toy example with a positional and semantic solution

## A.1   Phenomenology

In this appendix, we demonstrate that for a simple counting task two qualitatively different solutions exist in the loss landscape of a simple transformer using a dot-product attention layer with positional encodings – a practical example that further motivates our theoretical investigations in the main text. One solution corresponds to a dot-product attention matrix which is largely independent of the tokens making up the input sequence, and another strongly varies based on the tokens (and thus the semantic content of the input). Both solutions achieve a test accuracy close to 100%.

The training task is a sequence-to-sequence counting task, referred to as the *histogram task* in [6]. Given an input sequence $\boldsymbol{x} = [x_1, x_2, \cdots, x_L]$ of length $L$ of tokens from a fixed alphabet, the goal is to return a sequence $\boldsymbol{y} = [y_1, y_2, \cdots, y_L]$, where each token $y_i$ is the number of occurrences of the token $x_i$ in $\boldsymbol{x}$. In Fig. 1, we show an example where we consider sequences where the tokens are from the fixed alphabet $\mathcal{X} = \{A, B, C, \cdots\}$ of size $|\mathcal{X}| = 15$. In this setting, for instance, the sequence $\boldsymbol{x} = [A, B, B, C, A, B]$ should be mapped to its histogram sequence $\boldsymbol{y} = [2, 3, 3, 1, 2, 3]$. When the input data is limited to length $L$, the output elements $y_i$ thus take values up to the maximum count $L$.

We encode the input using token embeddings and absolute positional encodings which are trained jointly with the model weights. As an architecture we consider a small transformer made up of a single layer of dot-product attention, followed by a fully connected hidden layer and with learned embeddings for both tokens and positions. For each output position, it generates logits for the $L$ possible classes of the output alphabet; training is done using the cross-entropy loss. (Further details are provided in the remainder of this appendix).

We conduct experiments where we set two different sections of the model's weights to zero at the initialization of training –removing either the model's access to positional or semantic information– and keeping the weights frozen throughout training with the Adam optimizer. After convergence, we check that the resulting configurations of weights are stable in the unconstrained loss landscape, i.e.

without frozen weights. More precisely, we ascertain that these weights only change marginally when further trained with SGD on the unconstrained loss, and that the qualitative behaviour of the attention layer is retained. Our experiments demonstrate that the loss landscape of the transformer has at least two qualitatively different local minimizers (or close to minimizers), subsequently referred to as the semantic and positional solution.

We inspect the learnt attention matrix for different input sequences in Fig. 4. The positional solution corresponds to a learnt attention matrix whose $i, j-$th component only depend on the positions $i, j$, and little on the tokens occupying these positions. The attention matrix is thus almost independent of the input sequence. In fact, the attention matrix is similar to the identity. In this case, the attention layer simply serves to aggregate the other tokens uniformly, and the fully connected layer learns the counting.

In contrast, the attention matrix learnt at the semantic solution displays larger $i, j-$th component if the tokens at position $i$ and $j$ are identical. In other words, identical tokens attend more to each other. This mechanism hence does not rely on the positions, but rather on the semantic content of the tokens. Both solutions and associated attention matrices thus correspond to feasible algorithms which ultimately allow the transformer to solve the downstream task.

Our experimental exploration gives compelling evidence that different stable solutions exist in the empirical loss landscape of simple transformers, which correspond to different algorithmic solutions to a given task. However, it remains an interpretation of an experiment and does not allow for a precise characterization of their behaviour or of the conditions under which they are established. This stands in contrast to the example in the main paper: the model treated there is simpler, but still presents similar phenomenology and can be analyzed theoretically.

In the remainder of this Appendix we discuss the implementation details such as architecture and training procedure which enabled us to exhibit the different minima for the histogram task.

The code for reproducing the results is available at github.com/SPOC-group/positional-and-semantic-attention.

## A.2 Dataset

We use the histogram task as proposed in [6]. We consider sequences of fixed length $L = 15$. For every input sequence $\mathbf{x} = [x_1, x_2, \cdots, x_L]$ we sample $x_i$ i.i.d. and uniformly from a set of tokens $\mathcal{T}$ of size $T$, which we set to 15 in our experiments. For visualization purposes we use capitalized letters as tokens. To obtain the target $\mathbf{y} = [y_1, y_2, \cdots, y_L]$, we set $y_i = \sum_{j=1}^{L} \mathbb{1}(x_i = x_j)$, where $\mathbb{1}(b)$ is 1 if the boolean statement $b$ is true and zero otherwise. Therefore, $y_i \in \{1, 2, \cdots, L\}$.

## A.3 Model

In order to read the input sequence using a transformer we learn an embedding of dimension $d$ for each of the $T$ tokens and $L$ positions, stored in the matrices $E^{\text{token}} = [t_A, t_B, \cdots] \in \mathbb{R}^{T \times d}$ and $E^{\text{pos}} = [p_1, p_2, \cdots] \in \mathbb{R}^{L \times d}$. We convert the input sequence $\mathbf{x} = [x_1, x_2, \cdots, x_L] \in \mathcal{T}^L$ into an embedded sequence $\tilde{\mathbf{x}} \in \mathbb{R}^{L \times d}$ of the same length where $\tilde{x}_i = t_{x_i} + p_i$. This input sequence is then fed into the first layer of the transformer.

Formally, we have

$$\text{logits}_i(\mathbf{x}) = W_2 ReLU(W_1 LayerNorm(Attention_i(\tilde{\mathbf{x}})) + b_1) + b_2 \in \mathbb{R}^c \tag{19}$$

with $W_1 \in \mathbb{R}^{d \times h}, b_1 \in \mathbb{R}^h, W_2 \in \mathbb{R}^{h \times c}, b_2 \in \mathbb{R}^c$. The final prediction is obtained using the argmax on the logits. We have that the score matrix $A$ and the dot-product attention mechanism is

$$Attention(\mathbf{x}) = A(\mathbf{x})V\mathbf{x} \tag{20}$$

$$A(\mathbf{x}) = \text{Softmax}\left(\frac{\mathbf{x}QK^T\mathbf{x}^T}{\sqrt{d}}\right) \tag{21}$$

where $Q, K, V \in \mathbb{R}^{d \times d}$ and the softmax is applied row-wise. Also, for an $x \in \mathbb{R}^d$ we define

$$LayerNorm(x) = \frac{x - \mathbb{E}[x]}{\sqrt{Var(x) + \varepsilon}} * \gamma + \beta \tag{22}$$

where $\gamma, \beta \in \mathbb{R}^d$. We define the empirical loss for a dataset $\mathcal{D} = \{\mathbf{x}^\mu, \mathbf{y}^\mu\}_{\mu=1}^n$ as the average cross entropy loss for all output tokens $C$, i.e.

$$\mathcal{L}(\mathcal{D}) = \sum_{\mu=1}^{n} \sum_{l=1}^{L} - \sum_{c=1}^{C} [y_i^\mu]_c \log[\text{Softmax}(\text{logits}_i(\mathbf{x}^\mu))]_c . \tag{23}$$

Because of the way in which the histogram dataset is created for a fixed input length $L$, it follows $C = L$.

### A.4  Training procedure and freezing model parameters

To obtain the positional and semantic minima, we set some weights to zero in $E^{\text{token}}, E^{\text{pos}}, Q, K$ at initialization, and also freeze these zero weights during training. Note that this procedure was used in the literature to study architectural biases of varying model architectures, e.g. by [62] for convolutional neural networks.

With $\cdot$ we denote the initialization that is taken as the default `Pytorch` initialization for a linear layer. For both semantic and positional initialization, we overwrite this initialization with zeros as follows with $i = 1, \cdots, T$ and $j = 1, \cdots, L$

$$t_i = \left( \begin{array}{c|c} \cdot_{d/2} & \mathbf{0}_{d/2} \end{array} \right) , \; p_j = \left( \begin{array}{c|c} \mathbf{0}_{d/2} & \cdot_{d/2} \end{array} \right) , \tag{24}$$

where $\mathbf{0}_{d/2}$ is the all-zero vector of size $d/2$. For the *positional* initialization we additionally set

$$Q = \left( \begin{array}{c|c} \cdot_{d/2 \times d/2} & \mathbf{0}_{d/2 \times d/2} \\ \hline \mathbf{0}_{d/2 \times d/2} & \mathbf{0}_{d/2 \times d/2} \end{array} \right) , \; K = \left( \begin{array}{c|c} \cdot_{d/2 \times d/2} & \mathbf{0}_{d/2 \times d/2} \\ \hline \mathbf{0}_{d/2 \times d/2} & \mathbf{0}_{d/2 \times d/2} \end{array} \right) , \tag{25}$$

and for the *semantic* initialization we additionally set

$$Q = \left( \begin{array}{c|c} \mathbf{0}_{d/2 \times d/2} & \mathbf{0}_{d/2 \times d/2} \\ \hline \mathbf{0}_{d/2 \times d/2} & \cdot_{d/2 \times d/2} \end{array} \right) , \; K = \left( \begin{array}{c|c} \mathbf{0}_{d/2 \times d/2} & \mathbf{0}_{d/2 \times d/2} \\ \hline \mathbf{0}_{d/2 \times d/2} & \cdot_{d/2 \times d/2} \end{array} \right) , \tag{26}$$

where $\mathbf{0}_{d/2 \times d/2}$ is the all-zero matrix of size $d/2$ by $d/2$. In either case, the attention mechanism only has access to the semantic or positional part of the model.

Training all weights except the ones frozen to zero using Adam on a dataset of size $n = 35,000$, we obtain a $> 99.8\%$ test accuracy on a test set of size $n = 15,000$ for both datasets. We call these parameter configurations $\theta_{pos}$ and $\theta_{sem}$ respectively (the local minima as referred to in the main text).

In Fig. 4, we show that the attention layer of the two models behaves in qualitatively different ways, by showing the activations of the matrix $A$ for different input sequences.

### A.5  Unfreezing model parameters and SGD convergence

Finally, we want to verify that the parameter configurations $\theta_{pos}$ and $\theta_{sem}$ we obtained using the special initialization and frozen training are stable configurations in the unconstrained parameter space of the model. To show this, we perturb the given parameter configuration slightly using additive Gaussian i.i.d. noise with a scale of $0.001$. Subsequently, we run SGD on this perturbed configuration in the unconstrained parameter space for another 100 epochs. We call the parameter configurations obtained after this step $\tilde{\theta}_{pos}$ and $\tilde{\theta}_{sem}$.

While this further training leads the training loss to further decrease and the test accuracy to slightly increase, each models qualitative behaviour before and after extra training remains unchanged, as shown in Fig. 5 and 6, for the examples also used in Fig. 1. The absolute difference in norm between $\tilde{\theta}_{pos}$ and $\theta_{pos}$ (respectively $\tilde{\theta}_{sem}$ and $\theta_{sem}$) of parameters is also small (see Table 1).

This evidences, that both resulting parameter configurations are in flat regions of the full parameterization of the transformer model, and that these two regions are further qualitatively different in the same sense that the original ones were. We use this stability in the unconstrained loss landscape as a justification to refer to them as "local minima".

|  | $\theta^{\text{init}} \xrightarrow{\text{Adam}} \theta$ | | $\theta \xrightarrow{\text{SGD}} \tilde{\theta}$ |
|---|---|---|---|
|  | accuracy $\theta$ | $\|\theta^{\text{init}} - \theta\|_2$ | $\|\theta - \tilde{\theta}\|_2$ |
| positional | $0.99885 \pm 0.0004$ | $158.85 \pm 4.35$ | $0.61 \pm 0.13$ |
| semantic | $0.99895 \pm 0.0012$ | $163.69 \pm 8.88$ | $0.46 \pm 0.24$ |

Table 1: Parameter configurations for different types of initializations for the histogram task. Optimizer parameters as in the accompanying code. Average over 10 runs using the same dataset.

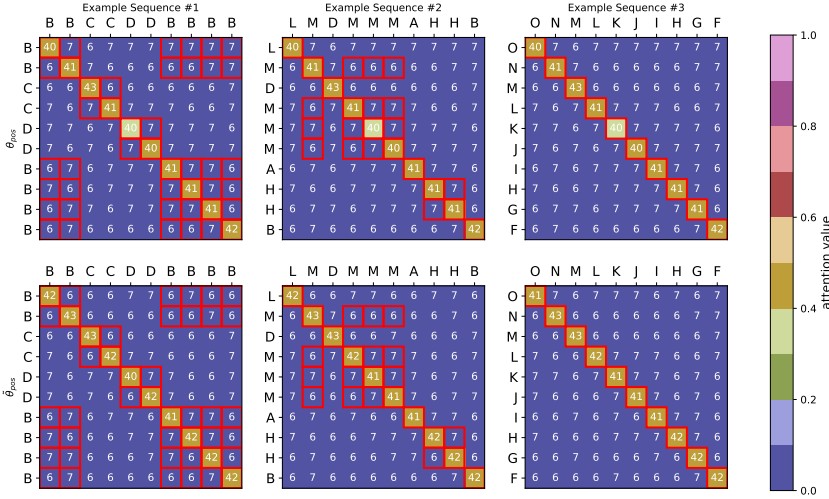

Figure 5: Comparison of the attention layer activations for different sequences for $\theta_{pos}$ and $\tilde{\theta}_{pos}$.

# B   Derivation of Result 4.2

In this Appendix, we provide a detailed derivation of Result 4.2 of the main text. In subsection B.2. we introduce a Generalized Approximate Message Passing algorithm (GAMP) [56]. Subsection B.3 then establishes that equations (7) of Result 4.2 track the dynamics of summary statistics describing the GAMP algorithm. In particular, the equations (7) describe the fixed points of GAMP. Finally, subsection B.4 shows that fixed points of GAMP correspond to critical (zero-gradient) points of the empirical loss landscape (3), thus establishing that equations 7 of Result 4.2 describe fixed points of GD.

## B.1   Notations

For simplicity, we place ourselves in the setting $r_s = 1$ explored in Section 5 of the main text, but allow the length $L$ of the sentences to be arbitrary, and allow a generic learning model S (2), i.e. not necessarily the dot-product attention model analyzed in Section 5. The case $r_s \geq 2$ follows identical derivation steps, modulo the replacement of all variables by tensor objects. We provide another alternative derivation of Result 4.2 in full generality in Appendix C, using the replica method from statistical physics. Let us note $\{X_\ell\}_{1 \leq \ell \leq L}$ a series of $L$ $n \times d$ matrices, with $X_\ell$ corresponding to the $\ell-$th rows (tokens) of each input sentence $x^\mu$ stacked vertically, and normalized by $\sqrt{d}$. We denote $\tilde{X}_\ell \equiv X_\ell + P_\ell$, where $P \in \mathbb{R}^{n \times d}$ is the matrix with all rows equal to the $\ell-$th positional encoding $p_\ell$. Let us further define $\rho \in \mathbb{R}^{n \times L \times L}$ the tensor corresponding to the sequence of $n$ matrices $\frac{1}{d} x^\mu (x^\mu)^\top \in \mathbb{R}^{L \times L}$. Finally, let us denote $T \in \mathbb{R}^{n \times L \times L}$ the tensor so that the $\mu-$th row of $T$ satisfies $y(x^\mu) = T^\mu x^\mu$, see equation (1). In other words, $T$ corresponds to the concatenation of the target attention matrices.

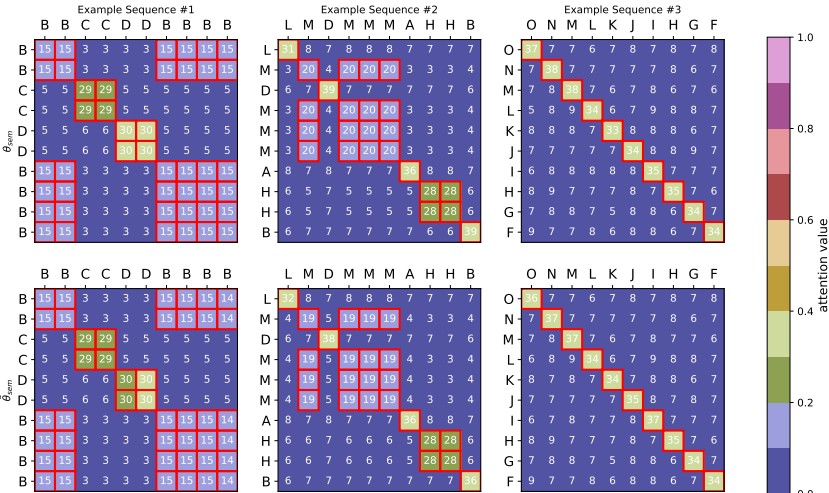

Figure 6: Comparison of the attention layer activations for different sequences for $\theta_{sem}$ and $\tilde{\theta}_{sem}$.

Before detailing the derivation, we first highlight a simplifying observation. Note that a loss item can be expanded as

$$\frac{1}{d}\left\|y(x^\mu) - \mathtt{S}\left[\frac{1}{\sqrt{d}}(x^\mu + p)Q\right]x^\mu\right\|^2 = \|y(x^\mu)\|^2 + \mathrm{Tr}\,\mathtt{S}\left[\frac{1}{\sqrt{d}}(x^\mu + p)Q\right]\rho_\Sigma \mathtt{S}\left[\frac{1}{\sqrt{d}}(x^\mu + p)Q\right]^\top$$

$$- 2\,\mathrm{Tr}\,\mathtt{T}\left[\frac{1}{\sqrt{d}}x_\ell Q_\star\right]\rho_\Sigma \mathtt{S}\left[\frac{1}{\sqrt{d}}(x_\ell^\mu + p_\ell)Q\right]^\top, \qquad (27)$$

where we used that with high probability in the considered asymptotic limit, for all $1 \le \mu \le n$,

$$xx^\top = (x+p)(x+p)^\top = x(x+p)^\top = \rho_\Sigma. \qquad (28)$$

Since the first term of (27) does not depend on the weights $Q$, it can be without loss of generality substracted from the loss. Without loss of generality, one can thus consider the equivalent empirical risk minimization problem

$$\hat{Q} = \underset{Q \in \mathbb{R}^{d \times r}}{\mathrm{argmin}} \sum_{\mu=1}^n \frac{1}{2d}\left[\mathrm{Tr}\,\mathtt{S}\left[\frac{1}{\sqrt{d}}(x^\mu+p)Q\right]\rho_\Sigma \mathtt{S}\left[\frac{1}{\sqrt{d}}(x^\mu+p)Q\right]^\top - 2\,\mathrm{Tr}\,\mathtt{T}\left[\frac{1}{\sqrt{d}}x_\ell Q_\star\right]\rho_\Sigma \mathtt{S}\left[\frac{1}{\sqrt{d}}(x_\ell^\mu+p_\ell)Q\right]^\top\right] + \frac{\lambda}{2}\|Q\|^2. \qquad (29)$$

The risks (29) and (3) are equivalent, and we shall use the former in the following.

Finally, for arguments $T \in \mathbb{R}^{L \times L}, \rho \in \mathbb{R}^{L \times L}, \omega \in \mathbb{R}^L, V \in \mathbb{R}^{L \times L}$ we introduce the resolvent

$$\mathrm{prox}(T, \rho, \omega, V) \equiv$$

$$\underset{x=\{x_\ell \in \mathbb{R}\}_{\ell=1}^L}{\mathrm{arginf}}\left\{\sum_{\ell,\kappa=1}^L (x_\ell - \omega_\ell)(V^{-1})_{\ell\kappa}(x_\kappa - \omega_\kappa) - 2\,\mathrm{Tr}\left[\mathtt{S}[x]\rho T^\top\right] + \mathrm{Tr}\left[\mathtt{S}[x]\rho \mathtt{S}[x]^\top\right]\right\} \qquad (30)$$

Note that the latter part of the bracketed term corresponds to the simplified loss (27) derived in the beginning of Appendix C, which is the one we shall without loss of generality consider in the present appendix. For ease of presentation, we place ourselves under Assumption 4.1, where all the input covariances $\{\Sigma_\ell\}_\ell$ are codiagonalizable. In the following, without loss of generality, we thus assume them diagonal, by placing ourselves in the common basis $\{e_i\}_{1 \le i \le d}$ of Assumption 4.1.

## B.2 AMP algorithm

We are now in a position to state the AMP algorithm:

---

**Algorithm 1** GAMP

---

**Inputs** : $\{\tilde{X}_\ell \in \mathbb{R}^{n \times d}\}_{\ell=1}^L, T \in \mathbb{R}^{n \times L \times L}, \rho \in \mathbb{R}^{n \times L \times L}$

**Initialize** $\hat{Q}^0 = \sim \mathcal{N}(0, \mathbb{I}_d), \hat{c}^0 = \mathbb{I}_d, \{f_\ell^0 = 0_n\}_{\ell=1}^L$

**for** $t \leq t_{\max}$ **do**

$\quad \forall 1 \leq \ell, \kappa \leq L, \quad V_{\ell\kappa}^t = (\tilde{X}_\ell \odot \tilde{X}_\kappa)\hat{c}^t$

$\quad \forall 1 \leq \ell \leq L, \quad \omega_\ell^t = \tilde{X}_\ell \hat{Q}^t - \sum_{\kappa=1}^L V_{\ell\kappa}^t f_\kappa^{t-1}$

$\quad \forall 1 \leq \ell \leq L, \quad f_\ell^t = \sum_\kappa (V^{-1})_{\ell\kappa} (\text{prox}(T, \rho, \omega^t, V^t)_\kappa - \omega_\kappa^t)$

$\quad \forall 1 \leq \ell, \kappa \leq L, \quad g_{\ell\kappa}^t = \partial_{\omega_\ell} f_\kappa^t$

$\quad A^t = - \sum_{\ell,\kappa=1}^L (\tilde{X}_\ell \odot \tilde{X}_\kappa)^\top g_{\ell\kappa}^t$

$\quad b^t = \sum_{\ell=1}^L \tilde{X}_\ell^\top f_\ell^t + A^t \odot \hat{Q}^t$

$\quad \hat{Q}^{t+1} = (\lambda \mathbb{I}_d + A^t)^{-1} b^t$

$\quad \hat{c}^{t+1} = (\lambda \mathbb{I}_d + A^t)^{-1}$

**end for**

**return** Estimator $\hat{Q}$

---

The GAMP algorithm can be derived in standard fashion from the Belief Propagation (BP) algorithm, see e.g. [56, 63] or [64] for an overview. Compared to the standard GAMP iterations for Generalized linear models, one needs to account for the fact that there exist different sources of data $X_\ell$ (corresponding to the $\ell-$ th tokens of each input sentence), and for the fact that the output of the equivalent GLM are $\mathbb{R}^{L \times L}$- valued attention matrices. In the following subsection, we show that the fixed points of GAMP 1 correspond to critical points of the empirical loss (3), i.e. fixed points of Gradient Descent (GD), allowing to connect Result 4.2 to our numerical experiments using GD.

### B.3  State evolution

In this section we show that the dynamics of the GAMP Algorithm 1 are tracked by the summary statistics of Result 4.2. In particular, the equations (7) describe the statistics of the GAMP fixed points. To see this, it is convenient to take as a starting point the relaxed Belief Propagation (rBP) equations, which are a step upstream in the derivation of the GAMP iterations, and which are asymptotically equivalent– see e.g. [64] for a review or e.g. [65], Appendix A, for a detailed walkthrough. The rBP equations read

As conventional, we note $\cdot_\mu$ the version of a variable $\cdot_{\mu \to i}$ where the summation also encompasses the index $i$, and $\cdot_i$ the version of a variable $\cdot_{i \to \mu}$ where the summation also encompasses the index $\mu$. Note that in all cases above the two variables differ by at most $\Theta_d(1/\sqrt{d})$.

**Concentration of** $(V_{\mu \to i}^t)_{\ell\kappa}$    We first study the statistics of $V_{\mu \to i}^t, A_{i \to \mu}^t$, remembering that the data $\tilde{x}_{\ell i}^\mu \equiv (x_\ell^\mu)_i/\sqrt{d} + (p_\ell)_i/\sqrt{d}$ in the notation of the main text, with $(x_\ell^\mu)_i = \Theta_d(1), \ (p_\ell)_i = \Theta_d(1/\sqrt{d})$. Replacing in the rBP updates:

$$
\begin{aligned}
(V_{\mu \to i}^t)_{\ell\kappa} &= \sum_{j \neq i} (\tilde{x}_{\ell j}^\mu)(\tilde{x}_{\kappa j}^\mu)\hat{c}_{j \to \mu}^t \\
&= \underbrace{\frac{1}{d} \sum_{j \neq i} (x_\ell^\mu)_j (x_\kappa^\mu)_j \hat{c}_{j \to \mu}^t}_{\delta_{\ell\kappa}\Theta_d(1) + (1-\delta_{\ell\kappa})\Theta_d(1/\sqrt{d})} + \underbrace{\frac{1}{d} \sum_{j \neq i} (x_\ell^\mu)_j (p_\kappa)_j \hat{c}_{j \to \mu}^t + (\ell \leftrightarrow \kappa)}_{\Theta_d(1/d)} + \underbrace{\frac{1}{d} \sum_{j \neq i} (p_\ell)_j (p_\kappa)_j \hat{c}_{j \to \mu}^t}_{\Theta_d(1/d)} \\
&= \delta_{\ell\kappa} \frac{1}{d} \sum_j (\Sigma_\ell)_{jj} \hat{c}_j^t \equiv V_\ell^t
\end{aligned}
\tag{31}
$$

**Algorithm 2** rBP

---

**Inputs** : $\{\tilde{X}_\ell \in \mathbb{R}^{n \times d}\}_{\ell=1}^L, T \in \mathbb{R}^{n \times L \times L}, \rho \in \mathbb{R}^{n \times L \times L}$

**Initialize** $\forall 1 \leq \mu \leq n, \ 1 \leq i \leq d, \ \hat{Q}_{i \to \mu}^0 = 0, \hat{c}_{i \to \mu}^0 = 1, \{f_{\ell \mu \to i}^0 = 0\}_{\ell=1}^L$

**for** $t \leq t_{\max}$ **do**

$\quad \forall 1 \leq \ell, \kappa \leq L, 1 \leq \mu \leq n, 1 \leq i \leq d, \ (V_{\mu \to i}^t)_{\ell \kappa} = \sum_{j \neq i} (\tilde{x}_{\ell j}^\mu)(\tilde{x}_{\kappa j}^\mu) \hat{c}_{j \to \mu}^t$

$\quad \forall 1 \leq \ell, 1 \leq \mu \leq n, 1 \leq i \leq d, \ \omega_{\ell, \mu \to i}^t = \sum_{j \neq i} \tilde{x}_{\ell, i}^\mu \hat{Q}_{j \to \mu}$

$\quad \forall 1 \leq \ell, 1 \leq \mu \leq n, 1 \leq i \leq d, \ f_{\ell, \mu \to i}^t = \sum_\kappa (V_{\mu \to i}^{-1})_{\ell \kappa}(\text{prox}(T_\mu, \rho_\mu, \omega_{\mu \to i}^t, V_{\mu \to i}^t)_\kappa - \omega_{\kappa, \mu \to i}^t)$

$\quad \forall 1 \leq \ell, \kappa \leq L, 1 \leq \mu \leq n, 1 \leq i \leq d, \ g_{\ell \kappa, \mu \to i}^t = \partial_{\omega_\ell} f_{\kappa \mu \to i}^t$

$\quad \forall 1 \leq \mu \leq n, 1 \leq i \leq d, A_{i \to \mu}^t = -\sum_{\ell, \kappa=1}^L \sum_{\nu \neq \mu} (\tilde{x}_{\ell i}^\nu)(\tilde{x}_{\kappa i}^\nu) g_{\ell \kappa, \nu \to i}^t$

$\quad \forall 1 \leq \mu \leq n, 1 \leq i \leq d, \ b_{i \to \mu}^t = \sum_{\ell=1}^L \sum_{\nu \neq \mu} x_{\ell i}^\nu f_{\ell, \nu \to i}^t$

$\quad \forall 1 \leq \mu \leq n, 1 \leq i \leq d, \ \hat{Q}_{i \to \mu}^{t+1} = (\lambda \mathbb{I}_d + A_{i \to \mu}^t)^{-1} b_{i \to \mu}^t$

$\quad \forall 1 \leq \mu \leq n, 1 \leq i \leq d, \ \hat{c}_{i \to \mu}^{t+1} = (\lambda \mathbb{I}_d + A_{i \to \mu}^t)^{-1}$

**end for**

---

**return** Estimator $\hat{Q}$

---

**Distribution of $\omega_{\ell, \mu \to i}^t$**     Let us first introduce the teacher local field

$$h_{\mu, \ell} = \sum_i (x_\ell^\mu)_i Q_i^\star. \tag{32}$$

Like e.g. [65], Appendix A, we first ascertain the joint distribution of $h_{\mu, \ell}, \omega_{\ell, \mu \to i}^t$ with respect to the data. These variables have mean

$$\mathbb{E}[\omega_{\ell, \mu \to i}^t] = \frac{p_\ell^\top \hat{Q}^t}{\sqrt{d}} \equiv m_\ell^t \tag{33}$$

and respective variance

$$\mathbb{V}[\omega_{\ell, \mu \to i}^t \omega_{\kappa, \nu \to j}^t] = \delta_{\mu\nu} \delta_{\ell\kappa} \frac{1}{d} \sum_{i,j} \hat{Q}_i^t (\Sigma_\ell)_{ij} \hat{Q}_j^t \equiv \delta_{\mu\nu} \delta_{\ell\kappa} q_\ell^t \tag{34}$$

$$\mathbb{E}[h_{\mu\ell} h_{\nu\kappa}] = \delta_{\mu\nu} \delta_{\ell\kappa} \frac{1}{d} \sum_{i,j} Q_i^\star (\Sigma_\ell)_{ij} Q_j^\star \equiv \delta_{\mu\nu} \delta_{\ell\kappa} \rho_\ell \tag{35}$$

$$\mathbb{E}[h_{\mu\ell} \omega_{\kappa, \nu \to j}^t] = \delta_{\mu\nu} \delta_{\ell\kappa} \frac{1}{d} \sum_{i,j} Q_i^\star (\Sigma_\ell)_{ij} \hat{Q}_j^t \equiv \delta_{\mu\nu} \delta_{\ell\kappa} \theta_\ell^t \tag{36}$$

**Distribution of $b_{i \to \mu}^t$**     Let us ascertain the distribution of $b_{i \to \mu}^t$.

$$b_{i \to \mu}^t = \sum_\ell \sum_{\nu \neq \mu} (\tilde{x}_\ell^\nu)_i \underbrace{(V_\ell^t)^{-1} \left(\text{prox}(T_\nu, \rho_\nu, \{\omega_{\kappa, \nu \to i}^t\}_\kappa, V_{\nu \to i}^t)_\ell - \omega_{\ell, \nu \to i}^t\right)}_{\equiv \tilde{\text{prox}}(T_\nu, \rho_\nu, \{\omega_{\kappa, \nu \to i}^t\}_\kappa, V_{\nu \to i}^t)_\ell}$$

$$= \sum_\ell \sum_{\nu \neq \mu} {}^1/\sqrt{d} ((x_\ell^\nu)_i + (p_\ell)_i) \Bigg[ \tilde{\text{prox}}(\mathtt{T}[\{h_{\nu \to i, \kappa}\}_\kappa], \rho_\nu, \{\omega_{\kappa, \nu \to i}^t\}_\kappa, V_{\nu \to i}^t)_\ell$$

$$+ {}^1/\sqrt{d} \sum_\gamma (x_\gamma^\nu)_i Q_i^\star \partial_{h_\gamma} \tilde{\text{prox}}(\mathtt{T}[\{h_{\nu \to i, \kappa}\}_\kappa], \rho_\nu, \{\omega_{\kappa, \nu \to i}^t\}_\kappa, V_{\nu \to i}^t)_\ell \Bigg], \tag{37}$$

leading asymptotically to

$$\mathbb{E}[b_{i\to\mu}^t] = \sum_\ell (\sqrt{d}p_\ell)_i \underbrace{\alpha\mathbb{E}_{H=\{h_\kappa\}\Xi=\{\xi_\kappa\}}\tilde{\text{prox}}(\text{T}[H], \rho_\Sigma, \{m_\kappa^t + \sqrt{q_\kappa^t}\xi_\kappa\}_\kappa, \{V_\kappa^t\}_\kappa)_\ell}_{\equiv \hat{m}_\ell^t}$$

$$+ Q_i^\star \sum_\ell (\Sigma_\ell)_{ii} \underbrace{\alpha\mathbb{E}_{H,\Xi}\partial_{h_\ell}\tilde{\text{prox}}(\text{T}[H], \rho_\Sigma, \{m_\kappa^t + \sqrt{q_\kappa^t}\xi_\kappa\}_\kappa, \{V_\kappa^t\}_\kappa)_\ell}_{\equiv \hat{\theta}_\ell^t} \qquad (38)$$

where the expectations bear over $\xi_\ell \sim \mathcal{N}(0,1)$ and $h_\ell \sim \mathcal{N}(\xi_\ell \theta_\ell^t/\sqrt{q_\ell^t}, \rho_\ell - (\theta_\ell^t)^2/q_\ell^t)$. The variance is given by

$$\mathbb{V}[b_i^t, b_j^t] = \delta_{ij} \sum_\ell (\Sigma_\ell)_{ii} \underbrace{\alpha\mathbb{E}_{H,\Xi}\tilde{\text{prox}}(\text{T}[H], \rho_\Sigma, \{m_\kappa^t + \sqrt{q_\kappa^t}\xi_\kappa\}_\kappa, \{V_\kappa^t\}_\kappa)_\ell^2}_{\equiv \hat{q}_\ell^t} \qquad (39)$$

**Concetration of $A_{i\to\mu}^t$**    Similarly to the derivation for $V_{\mu\to i}^t$, $A_{i\to\mu}^t$ concentrates to

$$A_{i\to\mu}^t = \sum_\ell \underbrace{-\alpha\frac{1}{V_\ell^t}\left(\mathbb{E}_{H,\Xi}\partial_{\omega_\ell}\text{prox}(\text{T}[H], \rho_\Sigma, \{m_\kappa^t + \sqrt{q_\kappa^t}\xi_\kappa\}_\kappa, \{V_\kappa^t\}_\kappa)_\ell - 1\right)}_{\equiv \hat{V}_\ell^t}(\Sigma_\ell)_{ii} \qquad (40)$$

**Recovering Result 4.2**    Wrapping up, we now massage these equations to recover equations (7) from Result 4.2 of the main text. Starting from (31):

$$V_\ell^t = \frac{1}{d}\sum_j (\Sigma_\ell)_{jj}\frac{1}{\lambda + \sum_\kappa \hat{V}_\kappa^{t-1}(\Sigma_\kappa)}$$

$$= \int d\nu(\gamma, \tau)\gamma_\ell \left(\lambda + \sum_\kappa \hat{V}_\kappa^{t-1}\gamma_\kappa\right)^{-1}. \qquad (41)$$

Next, for $q_\ell^t$ (34):

$$q_\ell^t = \frac{1}{d}\sum_i (\Sigma_\ell)_{ii}\left(\left(\sum_\kappa(\sqrt{d}(p_\kappa)_i\hat{m}_\kappa^{t-1} + Q_i^\star(\Sigma_\kappa)_{ii}\hat{\theta}_\kappa\right)^2 + (\Sigma_\kappa)_{ii}\hat{q}_\kappa^{t-1}\right)\left(\lambda + \sum_\kappa \hat{V}_\kappa^{t-1}(\Sigma_\kappa)\right)^{-2}$$

$$= \int d\nu(\gamma, \tau, \pi)\gamma_\ell\left(\left(\sum_\kappa \hat{m}_\kappa^{t-1}\tau_\kappa + \hat{\theta}_\kappa\gamma_\kappa\pi_\kappa\right)^2 + \gamma_\kappa\hat{q}_\kappa^{t-1}\right)\left(\lambda + \sum_\kappa \hat{V}_\kappa^{t-1}\gamma_\kappa\right)^{-2} \qquad (42)$$

For $\theta_\ell^t$ (34):

$$\theta_\ell^t = \frac{1}{d}\sum_i (\Sigma_\ell)_{ii}Q_i^\star\left(\sum_\kappa(\sqrt{d}(p_\kappa)_i\hat{m}_\kappa^{t-1} + Q_i^\star(\Sigma_\kappa)_{ii}\hat{\theta}_\kappa\right)\left(\lambda + \sum_\kappa \hat{V}_\kappa^{t-1}(\Sigma_\kappa)\right)^{-1} + o_d(1)$$

$$= \int d\nu(\gamma, \tau, \pi)\gamma_\ell\pi_\ell\left(\sum_\kappa \hat{m}_\kappa^{t-1}\tau_\kappa + \hat{\theta}_\kappa\gamma_\kappa\pi_\kappa\right)\left(\lambda + \sum_\kappa \hat{V}_\kappa^{t-1}\gamma_\kappa\right)^{-1}. \qquad (43)$$

Finally for $m_\ell^t$ (33):

$$m_\ell^t = \frac{1}{d}\sum_i (\sqrt{d}p_\ell)_i\left(\sum_\kappa(\sqrt{d}(p_\kappa)_i\hat{m}_\kappa^{t-1} + Q_i^\star(\Sigma_\kappa)_{ii}\hat{\theta}_\kappa\right)\left(\lambda + \sum_\kappa \hat{V}_\kappa^{t-1}(\Sigma_\kappa)\right)^{-1} + o_d(1)$$

$$= \int d\nu(\gamma, \tau, \pi)\tau_\ell\left(\sum_\kappa \hat{m}_\kappa^{t-1}\tau_\kappa + \hat{\theta}_\kappa\gamma_\kappa\pi_\kappa\right)\left(\lambda + \sum_\kappa \hat{V}_\kappa^{t-1}\gamma_\kappa\right)^{-1}. \qquad (44)$$

For $\hat{m}_\ell^t$ (38):

$$\hat{m}_\ell^t = \alpha\mathbb{E}_{H,\Xi}\frac{1}{V_\ell^t}\left[\text{prox}_\ell - \sqrt{q_\ell^t}\xi_\ell - m_\ell^t\right], \qquad (45)$$

while for $\hat{\theta}_\ell^t$ (38):

$$\hat{\theta}_\ell^t = \alpha \mathbb{E}_{H,\Xi} \frac{1}{V_\ell^t} \partial_{h_\ell} \left[ \text{prox}_\ell - \sqrt{q_\ell^t} \xi_\ell - m_\ell^t \right]$$

$$= \alpha \mathbb{E}_{H,\Xi} \frac{1}{V_\ell^t} \frac{h_\ell - \theta_\ell^t / \sqrt{q_\ell^t} \xi_\ell}{\rho_\ell - (\theta_\ell^t)^2 / q_\ell^t} \left[ \text{prox}_\ell - \sqrt{q_\ell^t} \xi_\ell - m_\ell^t \right]. \tag{46}$$

Now turning to $\hat{q}_\ell^t$:

$$\hat{q}_\ell^t = \alpha \mathbb{E}_{H,\Xi} \left[ \left( \frac{1}{V_\ell^t} \text{prox}_\ell - \sqrt{q_\ell^t} \xi_\ell - m_\ell^t \right)^2 \right]. \tag{47}$$

Finally, for $\hat{V}_\ell^t$ (40):

$$\hat{V}_\ell^t = -\alpha \mathbb{E}_{H,\Xi} \frac{1}{V_\ell^t} \left[ \partial_{\omega_\ell} \text{prox}_\ell - 1 \right]$$

$$= -\alpha \mathbb{E}_{H,\Xi} \frac{1}{V_\ell^t} \left[ \frac{1}{\sqrt{q_\ell^t}} \partial_\xi (\text{prox}_\ell - \sqrt{q_\ell^t} \xi_\ell - m_\ell) \right]$$

$$= \alpha \mathbb{E}_{H,\Xi} \frac{1}{\sqrt{q_\ell^t} V_\ell^t} \left[ \frac{\theta_\ell^t}{\sqrt{q_\ell^t} V_\ell^t} \left( \frac{h_\ell - \sqrt{q_\ell^t} \xi_\ell}{\rho_\ell - (\theta_\ell^t)^2 / q_\ell^t} - \xi \right) (\text{prox}_\ell - \sqrt{q_\ell^t} \xi_\ell - m_\ell) \right]$$

$$= \frac{\theta_\ell^t \hat{\theta}_\ell^t}{q_\ell^t} - \alpha \mathbb{E}_{H,\Xi} \frac{1}{\sqrt{q_\ell^t} V_\ell^t} (\text{prox}_\ell - \sqrt{q_\ell^t} \xi_\ell - m_\ell) \xi_\ell \tag{48}$$

**Summary : State evolution equations** The state evolution equations asymptotically describing the dynamics of the GAMP algorithm 1 thus read

$$\begin{cases} V_\ell^t = \int d\nu(\gamma, \tau) \gamma_\ell \left( \lambda + \sum_\kappa \hat{V}_\kappa^{t-1} \gamma_\kappa \right)^{-1} \\ q_\ell^t = \int d\nu(\gamma, \tau, \pi) \gamma_\ell \left( \left( \sum_\kappa \hat{m}_\kappa^{t-1} \tau_\kappa + \hat{\theta}_\kappa \gamma_\kappa \pi_\kappa \right)^2 + \gamma_\kappa \hat{q}_\kappa^{t-1} \right) \left( \lambda + \sum_\kappa \hat{V}_\kappa^{t-1} \gamma_\kappa \right)^{-2} \\ \theta_\ell^t = \int d\nu(\gamma, \tau, \pi) \gamma_\ell \pi_\ell \left( \sum_\kappa \hat{m}_\kappa^{t-1} \tau_\kappa + \hat{\theta}_\kappa \gamma_\kappa \pi_\kappa \right) \left( \lambda + \sum_\kappa \hat{V}_\kappa^{t-1} \gamma_\kappa \right)^{-1} \\ m_\ell^t = \int d\nu(\gamma, \tau, \pi) \tau_\ell \left( \sum_\kappa \hat{m}_\kappa^{t-1} \tau_\kappa + \hat{\theta}_\kappa \gamma_\kappa \pi_\kappa \right) \left( \lambda + \sum_\kappa \hat{V}_\kappa^{t-1} \gamma_\kappa \right)^{-1}. \end{cases} \tag{49}$$

$$\begin{cases} \hat{V}_\ell^t = \frac{\theta_\ell^t \hat{\theta}_\ell^t}{q_\ell^t} - \alpha \mathbb{E}_{H,\Xi} \frac{1}{\sqrt{q_\ell^t} V_\ell^t} (\text{prox}_\ell - \sqrt{q_\ell^t} \xi_\ell - m_\ell) \xi_\ell \\ \hat{q}_\ell^t = \alpha \mathbb{E}_{H,\Xi} \left[ \left( \frac{1}{V_\ell^t} \text{prox}_\ell - \sqrt{q_\ell^t} \xi_\ell - m_\ell^t \right)^2 \right] \\ \hat{\theta}_\ell^t = \alpha \mathbb{E}_{H,\Xi} \frac{1}{V_\ell^t} \frac{h_\ell - \theta_\ell^t / \sqrt{q_\ell^t} \xi_\ell}{\rho_\ell - (\theta_\ell^t)^2 / q_\ell^t} \left[ \text{prox}_\ell - \sqrt{q_\ell^t} \xi_\ell - m_\ell^t \right] \\ \hat{m}_\ell^t = \alpha \mathbb{E}_{H,\Xi} \frac{1}{V_\ell^t} \left[ \text{prox}_\ell - \sqrt{q_\ell^t} \xi_\ell - m_\ell^t \right] \end{cases} \tag{50}$$

which exactly recovers equations (7) of Result 4.2 of the main text, for the case $r_s = 1$ considered in the present Appendix. Again, we mention that the case $r_s \geq 2$ should follow straightforwardly with the exact same derivation steps, using tensor variables (see e.g. [52]). This subsection has thus established that the equations (7) (with time indices) describe the summary statistics capturing the dynamics of GAMP iterations 1. In particular, (7) describe the fixed points of GAMP. The next subsection further shows that the (stable) fixed points of GAMP correspond to critical (zero-gradient) points of the empirical landscape (3), i.e. fixed points of gradient descent. Finally, we provide in Appendix C an alternative derivation of the state evolution equations (7)(49), using the replica method from statistical physics [61, 66].

## B.4 Fixed points of GAMP are fixed points of GD

In this subsection, we show that fixed points of GAMP 1, as asymptotically described by (7) in Result 4.2, correspond to critical (zero gradient) points of the empirical landscape (3). Again, we present the

result for $r_s = 1$ for clarity, the generalization to $r_s \geq 2$ being straightforward (see e.g. [52]). In the previous notations, let us denote the (simplified, see (27)) empirical loss as

$$L(\{\tilde{X}_\ell Q\}_\ell) + g(Q) \tag{51}$$

where we introduced the shorthands

$$L(\{h_\ell\} \in \mathbb{R}^n) \equiv \sum_{\mu=1}^{n} \left( -2\operatorname{Tr}\left[\mathsf{S}[\{h_\ell^\mu\}_\ell]\rho_\mu T_\mu^\top\right] + \operatorname{Tr}\left[\mathsf{S}[\{h_\ell^\mu\}_\ell]\rho_\mu \mathsf{S}[\{h_\ell^\mu\}_\ell]^\top\right] \right) \tag{52}$$

$$g(Q) \equiv \frac{\lambda}{2}\|Q\|^2, \tag{53}$$

i.e. respectively the simplified empirical loss (3) and the regularization, as functions with matrix arguments. The empirical minimization problem (3) can thus be written compactly as

$$\hat{Q} = \operatorname*{argmin}_{Q \in \mathbb{R}^d} \left\{ L(\{\tilde{X}_\ell Q\}_\ell) + r(Q) \right\} \tag{54}$$

with the critical (zero-gradient) condition being given by

$$\sum_{\ell=1}^{L} \tilde{X}_\ell^\top \partial_\ell L(\{\tilde{X}_\ell Q\}_\ell) + \partial g(Q) \overset{!}{=} 0. \tag{55}$$

Let us choose a diagonal definite $A \in \mathbb{R}^{d \times d}$, and a sequence $\{V_\mu\}_{1 \leq \mu \leq n}$ of symmetric definite $L \times L$ matrices. Group them into a block diagonal matrix $\check{V} \in \mathbb{R}^{Ln \times Ln}$, so that the $\mu-$ th block of $\check{V}$ corresponds to $V_\mu$. It shall prove useful to further introduce the matrices $\check{X} \in \mathbb{R}^{d \times nL}$ (resp. $\check{\partial}L(\check{X}) \in \mathbb{R}^{nL}$), defined as the concatenation of the matrices $\tilde{X}_1, ..., \tilde{X}_L$ (resp. $\partial_1 L, ..., \partial_L L$), viewed as $n$ blocks of length $L$. Then without loss of generality the zero-gradient condition can be rewritten as

$$\check{X}^\top V^{-1}\left(V\check{\partial}L(\check{X}) - \check{X}Q\right) + A(A^{-1}\partial g(Q) + Q) \overset{!}{=} \check{X}^\top V^{-1}\check{X}Q + AQ. \tag{56}$$

Similarly to [52], let us introduce

$$\check{\omega} \equiv V\check{\partial}L(\check{X}) - \check{X}Q. \tag{57}$$

This can be written in terms of a resolvent as

$$\check{X}Q = \operatorname{prox}(\check{\omega}) \tag{58}$$

where

$$\operatorname{prox}(\check{\omega}) \in \mathbb{R}^{nL} = \operatorname*{argmin}_{\check{x} \in \mathbb{R}^{nL}} \left\{ \frac{1}{2}\|\check{x} - \check{\omega}\|_V^2 + L(\check{x}) \right\} \tag{59}$$

which corresponds to (30). Similarly, we denote

$$b \equiv A^{-1}\partial g(Q) + Q \tag{60}$$

So that

$$Q = \operatorname{prox}_g(b) = \operatorname*{argmin}_{x \in \mathbb{R}^d} \left\{ \frac{1}{2}\|x - b\|_{A^{-1}}^2 + g(x) \right\} \tag{61}$$

In the particular case of an $\ell_2$ regularization $g(\cdot) = \lambda/2\|\cdot\|^2$, note that

$$\operatorname{prox}_g(b) = (\lambda\mathbb{I}_d + A)^{-1}Ab. \tag{62}$$

The zero-gradient condition can now be rewritten as

$$\begin{cases} \check{X}^\top V^{-1}\left(\operatorname{prox}(\check{\omega}) - \check{\omega}\right) = A(b - \operatorname{prox}_g(b)) \\ \check{X}\operatorname{prox}_g(b) = \operatorname{prox}(\check{\omega}) \end{cases} \tag{63}$$

One is now in a position to expand the concatenated variables $\breve{\cdot}$ into a sequence of $L$ $n-$dimensional parameters. For $u = \text{prox}(\breve{\omega}), \breve{\omega}$ let us denote $u_{\mu\ell}$ $(1 \le \mu \le n, 1 \le \ell \le L)$ the $\ell-$th component of the $\mu-$th block. Introduce

$$f_{\mu\ell} \equiv \sum_{\kappa} (V_\mu^{-1})_{\ell\kappa} (\text{prox}(\breve{\omega})_{\mu\kappa} - \breve{\omega}_{\mu\kappa}). \tag{64}$$

Denote $f_\ell \equiv (f_{\mu\ell})_{1 \le \mu \le n} \in \mathbb{R}^n, \omega_\ell \equiv (\omega_{\mu\ell})_{1 \le \mu \le n} \in \mathbb{R}^n$. The system of equations (63) can then be rewritten as (further redefining $b \leftarrow Ab$):

$$\begin{cases} \sum_\ell \tilde{X}_\ell^\top f_\ell = b - A(\lambda \mathbb{I}_d + A)^{-1} b \\ \tilde{X}_\ell (\lambda \mathbb{I}_d + A)^{-1} b = \sum_\kappa V_{\ell\kappa} f_\kappa - \omega_\ell \end{cases} \tag{65}$$

We used the assumption that $g(\cdot)$ is an $\ell_2$ regularization. Finally, introducing $\hat{Q} = \text{prox}_g(A^{-1}b) = (\lambda \mathbb{I}_d + A)^{-1} b$, on reaches

$$\begin{cases} \sum_\ell \tilde{X}_\ell^\top f_\ell = b - A\hat{Q} \\ \tilde{X}_\ell \hat{Q} = \sum_\kappa V_{\ell\kappa} f_\kappa - \omega_\ell \end{cases} \tag{66}$$

which corresponds to the fixed-point equations of GAMP (Algorithm 1). This finishes to show the correspondence between the fixed points of GAMP and the critical points of the empirical landscape (3). To summarize, we have shown that equations (7) describe the zero-gradient points of the empirical loss landscape (3), i.e. fixed points of GD.

### B.5 Towards a rigorous proof of result 4.2

While the connection between the GAMP fixed point and the extrema of the loss is sound, and has been at the roots of many rigorous results for convex losses, see e.g. [57, 58, 59, 60, 48], there exist technical difficulties in adapting these rigorous arguments to the present setting, and a fully rigorous proof would warrant sizable work. While we leave this challenging task for future work, we wish to discuss how it can be potentially achieved. The first task would require the proof of point-wise convergence of GAMP, as indeed, the identification of the GAMP estimates with the one of the extrema of the loss function requires to be at the fixed point of the iteration. This difficulty, discussed in detail in, e.g [67, 60, 48], can be in principle addressed by computing the convergence criterion from the state evolution equations (see [67] the discussion in Lemma 7 in [60]), a criterion sometimes called the "replicon" in the context of replica theory [68].

Provided the replicon criterion is satisfied, all converging fixed point described by our theory thus correspond rigorously to fixed point of the loss. The last task would be to prove that the minimum of the loss is indeed the fixed point we found with minimum energy. A potential strategy to prove this would be to use the Gordon-Minimax approach of [69]. While it is used in many situations for convex problems (e.g. [70, 71, 72]), only one side would be required for our (non-convex) problem thanks to the GAMP matching bound. We hope that our results would provide inspiration for further research in this direction.

## C Derivation of Result 4.2 with the replica method

In the Appendix we provide an alternative derivation of Result 4.2, which sharply characterizes the global minimum of the empirical loss (3), using the heuristic replica method from statistical physics [61, 66] in its replica-symmetric formulation. First observe that for any test function $\phi(\hat{Q})$ of the minimizer $\hat{Q}$ of (3),

$$\phi(\hat{Q}) = \lim_{\beta \to \infty} \mathbb{E}_\mathcal{D} \frac{1}{Z} \int dQ \phi(Q) e^{-\beta \mathcal{R}[Q]}, \tag{67}$$

where we denoted $R[Q]$ the empirical loss (3), and

$$Z \equiv \int dQ e^{-\beta \mathcal{R}[Q]} \tag{68}$$

the normalization factor, also known as the *partition function* in statistical physics. We remind that $\mathcal{D}$ refers to the training set. In order to access key summary statistics and learning metrics associated to $\hat{Q}$, it is therefore reasonable to seek to compute the generating function associated to the measure (67), namely $\mathbb{E}\ln Z$. Such computations can be addressed using the *replica* method from statistical physics [61, 66], building on the identity

$$\ln Z = \lim_{s \to 0} \frac{Z^s - 1}{s}.$$

(69)

The backbone of the derivation thus lies in the computation of $\mathbb{E}Z^s$. Below, we detail the derivation for a generic convex regularizer $g : \mathbb{R}^d \to \mathbb{R}_+$ and later specialize to the case of $\ell_2$ regularization. The replicated partition function thus reads

$$
\mathbb{E}Z^s = \int \prod_{a=1}^{s} dQ_a e^{-\beta \sum_{a=1}^{s} g(Q_a)}
$$
$$
\prod_{\mu=1}^{n} \mathbb{E}_x e^{-\beta \sum_{a=1}^{s} \left( \mathrm{Tr}\, \mathtt{S}\left[\frac{1}{\sqrt{d}}(x+p)Q_a\right] \rho_\Sigma \mathtt{S}\left[\frac{1}{\sqrt{d}}(x+p)Q_a\right]^\top - 2\,\mathrm{Tr}\, \mathtt{T}\left[\frac{1}{\sqrt{d}}x_\ell Q_\star\right] \rho_\Sigma \mathtt{S}\left[\frac{1}{\sqrt{d}}(x+p)Q_a\right]^\top \right)}.
$$

(70)

Introduce the local fields

$$
h^a \equiv \frac{xQ_a}{\sqrt{d}} \in \mathbb{R}^{L \times r}, \qquad\qquad h^\star \equiv \frac{xQ_\star}{\sqrt{d}} \in \mathbb{R}^{L \times t}
$$

(71)

and the overlaps

$$
m_a \equiv \frac{pQ_a}{\sqrt{d}} \in \mathbb{R}^{L \times r},
$$

(72)

with rows $m_a^\ell$. These fields have statistics

$$
\mathbb{E}_x[h_\ell^a (h_\kappa^b)^\top] = \delta_{\ell\kappa} \frac{Q_a^\top \Sigma_\ell Q_b}{d} \equiv q_{ab}^\ell
$$

(73)

$$
\mathbb{E}_x[h_\ell^\star (h_\kappa^\star)^\top] = \delta_{\ell\kappa} \frac{Q_\star^\top \Sigma_\ell Q_\star}{d} \equiv \rho_\ell
$$

(74)

$$
\mathbb{E}_x[h_\ell^a (h_\kappa^\star)^\top] = \delta_{\ell\kappa} \frac{Q_a^\top \Sigma_\ell Q_\star}{d} \equiv \theta_a^\ell.
$$

(75)

Thus

$$
\mathbb{E}Z^s = \int dm d\hat{m} d\theta d\hat{\theta} dq d\hat{q}\, \underbrace{e^{-d \sum_a \sum_\ell [\hat{m}_a^{\ell\top} m_a^\ell + \mathrm{Tr}(\theta_a^\ell \hat{\theta}_a^{\ell\top})] - d \sum_\ell \sum_{1 \le a \le b \le s} \mathrm{Tr}(q_{ab}^\ell \hat{q}_{ab}^{\ell\top})}}_{e^{sd\Psi_t}}
$$
$$
\underbrace{\int \prod_{a=1}^{s} dQ_a e^{-\beta \sum_{a=1}^{s} g(Q_a) + \sum_a \sum_\ell (\sqrt{d}\hat{m}_a^{\ell\top} Q_a^\top p_\ell + \mathrm{Tr}[\theta_a^\ell Q_\star^\top \Sigma_\ell Q_a]) + \sum_{1 \le a \le b \le s} \sum_\ell \mathrm{Tr}[q_{ab}^\ell Q_b^\top \Sigma_\ell Q_a]}}_{e^{sd\Psi_Q}}
$$
$$
\underbrace{\left[ \mathbb{E}_{h^\star, \{h_a\}_{a=1}^s} e^{-\beta \sum_{a=1}^{s} \left( \mathrm{Tr}\, \mathtt{S}[h^a + m^a] \rho_\Sigma \mathtt{S}[h^a + m^a]^\top - 2\,\mathrm{Tr}\, \mathtt{T}[h^\star] \rho_\Sigma \mathtt{S}[h^a + m^a]^\top \right)} \right]^{\alpha d}}_{e^{s\alpha d\Psi_y}},
$$

(76)

where we decomposed the replicated free entropy into the trace, entropic and energetic potentials $\Psi_t, \Psi_Q, \Psi_y$. Note that all exponents are scaling with $d \to \infty$. Therefore the integral in (76) can be computed using a Laplace saddle-point approximation.

## C.1 Replica-Symmetric ansatz

We have thus rephrased the analysis of the measure (67) as a optimization problem over the order parameters $\{q_{ab}^\ell, \theta_a^\ell, m_a\}$, and the associated conjugate variables. However, these still represent

$2L(s^2 + 1) + 2s$ variables, and $s \to 0$. In order to make progress, we assume that the maximizer is of *replica-symmetric* (RS) form [66, 61]

$$q_{ab}^\ell = (r_\ell - q_\ell)\delta_{ab} + q_\ell \tag{77}$$

$$m_a^\ell = m_\ell \tag{78}$$

$$\theta_a^\ell = \theta_\ell \tag{79}$$

$$\hat{q}_{ab}^\ell = -(\hat{r}_\ell/2 + \hat{q}_\ell) + \hat{q}_\ell \tag{80}$$

$$\hat{m}_a^\ell = \hat{m}_\ell \tag{81}$$

$$\hat{\theta}_a^\ell = \hat{\theta}_\ell \tag{82}$$

The RS ansatz holds in a number of machine learning settings, notably for convex problems and Bayes-optimal settings, see e.g. [64] for a review. In the present setting, since the empirical loss (3) is non-convex, we emphasize that the RS ansatz constitutes a heuristic technical assumption of our analysis.

## C.2 Entropic potential

We now turn to the entropic potential $\Psi_w$. It is convenient to introduce the variance order parameter

$$\hat{V}_\ell \equiv \hat{r}_\ell + \hat{q}_\ell. \tag{83}$$

The entropic potential can then be expressed as

$e^{\beta s d \Psi_Q}$

$$= \int \prod_{a=1}^s dQ_a e^{-\beta \sum_a g(Q^a) + \sum_{\ell=1}^L \sum_{a=1}^s \left( \sqrt{d}\hat{m}_\ell^\top Q_a^\top p_\ell + \mathrm{Tr}\left[\hat{Q}_\star^\top \Sigma_\ell Q_a\right]\right) - \frac{1}{2}\sum_{\ell=1}^L \sum_{a=1}^s \mathrm{Tr}\left[\hat{V}_\ell Q_a \Sigma_\ell Q_a^\top\right] + \frac{1}{2}\sum_{\ell=1}^L \sum_{a,b} \mathrm{Tr}\left[\hat{q}_\ell Q_a \Sigma_\ell Q_b^\top\right]}$$

$$= \int \prod_{\ell=1}^L D\boldsymbol{\Xi}_\ell$$

$$\left[ \int dQe^{-\beta g(Q) - \frac{1}{2}\mathrm{Tr}\left[\sum_{\ell=1}^L \hat{V}_\ell Q \boldsymbol{\Sigma}_\ell Q^\top\right] + \left(\sum_{\ell=1}^L \left(\sqrt{d}\hat{m}_\ell p_\ell^\top + \hat{\theta}_\ell Q_\star^\top \Sigma_\ell\right) + \sum_{\ell=1}^L \boldsymbol{\Xi}_\ell \odot (\hat{q}_k \otimes \boldsymbol{\Sigma}_k)^{\frac{1}{2}}\right) \odot Q} \right]^s$$

$$= \mathbb{E}_{\Xi}\left[ \int dQe^{-\beta g(Q) - \frac{1}{2}Q \odot \left[\sum_{\ell=1}^L \hat{V}_\ell \otimes \boldsymbol{\Sigma}_\ell\right] \odot Q + \left(\sum_{\ell=1}^L \left(\sqrt{d}\hat{m}_\ell p_\ell^\top + \hat{\theta}_\ell Q_\star^\top \Sigma_\ell\right) + \sum_{\ell=1}^L \boldsymbol{\Xi}_\ell \odot (\hat{q}_\ell \otimes \boldsymbol{\Sigma}_\ell)^{\frac{1}{2}}\right) \odot Q} \right]^s.$$

$$\tag{84}$$

Therefore

$$\beta \Psi_w = \frac{1}{d}\int \mathbb{E}_{\Xi} \ln\left[ \int dQe^{-\beta g(Q) - \frac{1}{2}Q \odot \left[\sum_{\ell=1}^L \hat{V}_\ell \otimes \boldsymbol{\Sigma}_\ell\right] \odot Q + \left(\sum_{\ell=1}^L \left(\sqrt{d}\hat{m}_\ell p_\ell^\top + \hat{\theta}_\ell Q_\star^\top \Sigma_\ell\right) + \sum_{\ell=1}^L \boldsymbol{\Xi}_\ell \odot (\hat{q}_\ell \otimes \boldsymbol{\Sigma}_\ell)^{\frac{1}{2}}\right) \odot Q} \right].$$

$$\tag{85}$$

For a matrix $\boldsymbol{\Xi} \in \mathbb{R}^{r \times d}$ and tensors $\boldsymbol{A}, \boldsymbol{B} \in \mathbb{R}^{r \times d} \otimes \mathbb{R}^{r \times d}$, we denoted $(\boldsymbol{\Xi} \odot \boldsymbol{A})_{kl} = \sum_{ij} \boldsymbol{\Xi}^{ij} \boldsymbol{A}_{ij,kl}$ and $(\boldsymbol{A} \odot \boldsymbol{B})_{ij,kl} = \sum_{rs} \boldsymbol{A}_{ij,rs} \boldsymbol{B}_{rs,kl}$.

## C.3 Energetic potential

The computation of the energetic potential $\Psi_y$ is rather standard and follows the same lines as in e.g. [20], yielding

$$\beta\Psi_y = \int_{\mathbb{R}^{L\times t}} dY\,DZ \int_{\mathbb{R}^{L\times r}} D\Xi \prod_{\ell=1}^L \delta\left[ y_\ell - (\rho_\ell - \theta_\ell^\top q_\ell^{-1}\theta_\ell)^{\frac{1}{2}} z_\ell - \theta_\ell^\top q_\ell^{\frac{1}{2}}\xi_\ell \right]$$

$$\times \ln\left[ \int_{\mathbb{R}^{L\times r}} dX \prod_{\ell=1}^L \frac{e^{-\frac{1}{2}\left(x_\ell - q_\ell^{\frac{1}{2}}\xi_\ell\right)^\top V_\ell^{-1}\left(x_\ell - q_\ell^{\frac{1}{2}}\xi_\ell\right)}}{\det(2\pi V_\ell)} e^{-\beta\operatorname{Tr}\mathsf{S}[x+m]\rho_\sigma \mathsf{S}[x+m]^\top - 2\operatorname{Tr}\mathsf{T}[y]\rho_\Sigma \mathsf{S}[x+m]^\top} \right]$$

$$= \underbrace{\int_{\mathbb{R}^{L\times t}} dY \int_{\mathbb{R}^{L\times r}} D\Xi \prod_{\ell=1}^L \frac{e^{-\frac{1}{2}\left(y_\ell - \theta_\ell^\top q_\ell^{\frac{1}{2}}\xi_\ell\right)^\top (\rho_\ell - \theta_\ell^\top q_\ell^{-1}\theta_\ell)^{-1}\left(y_\ell - \theta_\ell^\top q_\ell^{\frac{1}{2}}\xi_\ell\right)}}{\det\left[2\pi(\rho_\ell - \theta_\ell^\top q_\ell^{-1}\theta_\ell)\right]}}_{\equiv\,\mathbb{E}_{Y,\Xi}}$$

$$\times \ln\left[ \int_{\mathbb{R}^{L\times r}} dX \prod_{\ell=1}^L \frac{e^{-\frac{1}{2}\left(x_\ell - q_\ell^{\frac{1}{2}}\xi_\ell\right)^\top V_\ell^{-1}\left(x_\ell - q_\ell^{\frac{1}{2}}\xi_\ell\right)}}{\det(2\pi V_\ell)} e^{-\beta\operatorname{Tr}\mathsf{S}[x+m]\rho_\sigma \mathsf{S}[x+m]^\top - 2\operatorname{Tr}\mathsf{T}[y]\rho_\Sigma \mathsf{S}[x+m]^\top} \right]$$

$$\tag{86}$$

## C.4 Zero-temperature limit

We now take the limit $\beta \to \infty$. Rescaling

$$\beta\hat{V}_\ell \leftarrow \hat{V}_\ell, \qquad \frac{1}{\beta}V_\ell \leftarrow V_\ell, \qquad \beta\hat{m}_\ell \leftarrow \hat{m}_\ell, \qquad \beta\hat{\theta}_\ell \leftarrow \hat{\theta}_\ell, \qquad \beta^2\hat{q}_\ell \leftarrow \hat{q}_\ell \tag{87}$$

The entropic potential then reduces to

$$\Psi_w = \frac{1}{2d}\mathbb{E}_\Xi \operatorname{Tr}\left[ \left(\sum_{\ell=1}^L \hat{V}_\ell \otimes \Sigma_\ell\right) \odot \left( \sum_{\ell=1}^L \left(\sqrt{d}\hat{m}_\ell p_\ell^\top + \hat{\theta}_\ell Q_\star^\top \Sigma_\ell\right) + \sum_{\ell=1}^L \Xi_\ell \odot (\hat{q}_\ell \otimes \Sigma_\ell)^{\frac{1}{2}} \right)^{\otimes 2} \right]$$

$$- \frac{1}{d}\mathbb{E}_\Xi \mathcal{M}_g(\Xi) \tag{88}$$

where we defined the entropic Moreau enveloppe

$$M_g(\Xi) \equiv \inf_Q \left\{ \frac{1}{2}\left\| \left(\textstyle\sum_{\ell=1}^L \hat{V}_\ell \otimes \Sigma_\ell\right)^{1/2} \left(Q - \left(\textstyle\sum_{\ell=1}^L \hat{V}_\ell \otimes \Sigma_\ell\right)^{-1}\left(\textstyle\sum_{\ell=1}^L (\sqrt{d}\hat{m}_\ell p_\ell^\top + \hat{\theta}_\ell Q_\star^\top \Sigma_\ell) + \textstyle\sum_{\ell=1}^L \Xi_\ell \odot (\hat{q}_\ell \otimes \Sigma_\ell)^{\frac{1}{2}}\right)\right) \right\|^2 + g(Q) \right\}. \tag{89}$$

The energetic potential can be similarly recast into a more compact form

$$\Psi_y = -\mathbb{E}_{Y,\Xi}\mathcal{M}(Y,\Xi) \tag{90}$$

where the Moreau envelope is defined as

$$\mathcal{M}(Y,\Xi) = \inf_X \frac{1}{2}\left\{ \sum_{\ell=1}^L \operatorname{Tr}\left[ V_\ell^{-1}\left(x_\ell - q_\ell^{1/2}\xi_\ell - m_\ell\right)^{\otimes 2} \right] + \operatorname{Tr}\left[\mathsf{S}(X)\rho_\Sigma \mathsf{S}(X)^\top\right] - 2\operatorname{Tr}\left[\mathsf{T}(Y)\rho_\Sigma \mathsf{S}(X)^\top\right] \right\}. \tag{91}$$

## C.5 Replica free entropy

One finally reaches an expression for the replica free entropy as

$$\Phi = \frac{1}{2}\sum_{\ell=1}^L \left(\operatorname{Tr}\hat{V}_\ell q_\ell - \operatorname{Tr}\hat{q}_\ell V_\ell\right) - \sum_{\ell=1}^L \hat{m}_\ell^\top m_\ell - \sum_{\ell=1}^L \operatorname{Tr}\hat{\theta}_\ell^\top \theta_\ell - \frac{1}{d}\mathbb{E}_\Xi \mathcal{M}_g(\Xi)$$

$$+ \frac{1}{2d}\mathbb{E}_\Xi \operatorname{Tr}\left[ \left(\sum_{\ell=1}^L \hat{V}_\ell \otimes \Sigma_\ell\right) \odot \left( \left(\sqrt{d}\hat{m}_\ell p_\ell^\top + \hat{\theta}_\ell Q_\star^\top \Sigma_\ell\right) + \sum_{\ell=1}^L \Xi_\ell \odot (\hat{q}_\ell \otimes \Sigma_\ell)^{\frac{1}{2}} \right)^{\otimes 2} \right] - \alpha\mathbb{E}_{y,\xi}\mathcal{M}(y,\xi)$$

$$\tag{92}$$

## C.6 Saddle-point equations : general regularizer

The extremization of the free entropy (92) yields, similarly to [47], the following system of self-consistent equations on the summary statistics:

$$
\begin{cases}
V_\ell = \frac{1}{d}\mathbb{E}_{\boldsymbol{\Xi}}\left[\left(\mathrm{prox}_g \odot (\hat{q}_\ell \otimes \boldsymbol{\Sigma}_\ell)^{-\frac{1}{2}} \odot (\mathbb{I}_r \otimes \boldsymbol{\Sigma}_\ell)\right)\boldsymbol{\Xi}_\ell^\top\right] \\
q_\ell = \frac{1}{d}\mathbb{E}_{\boldsymbol{\Xi}}\left[\mathrm{prox}_g \boldsymbol{\Sigma}_\ell \mathrm{prox}_g^\top\right] \\
m_\ell = \frac{1}{\sqrt{d}}\mathbb{E}_{\boldsymbol{\Xi}}\left[\mathrm{prox}_g p_\ell\right] \\
\theta_\ell = \frac{1}{\sqrt{d}}\mathbb{E}_{\boldsymbol{\Xi}}\left[\mathrm{prox}_g \Sigma_\ell Q_\star\right]
\end{cases}
\tag{93}
$$

$$
\begin{cases}
\hat{q}_\ell = \alpha\mathbb{E}_{\boldsymbol{\Xi},Y}V_\ell^{-1}\left(\mathrm{prox}_\ell - q_\ell^{\frac{1}{2}}\xi_\ell - m_\ell\right)^{\otimes 2}V_\ell^{-1} \\
\hat{V}_\ell = \hat{\theta}_\ell\theta_\ell^\top q_\ell^{-1} - \alpha\mathbb{E}_{\boldsymbol{\Xi},Y}V_\ell^{-1}\left(\mathrm{prox}_\ell - q_\ell^{\frac{1}{2}}\xi_\ell - m_\ell\right)\xi_\ell^\top q_\ell^{-\frac{1}{2}} \\
\hat{m}_\ell = \alpha\mathbb{E}_{\xi,\eta}V_\ell^{-1}\left(\mathrm{prox}_\ell - q_\ell^{\frac{1}{2}}\xi_\ell - m_\ell\right) \\
\hat{\theta}_\ell = \alpha\mathbb{E}_{\xi,\eta}V_\ell^{-1}\left(\mathrm{prox}_\ell - q_\ell^{\frac{1}{2}}\xi_\ell - m_\ell\right)\left(y_\ell - \xi_\ell^\top q_\ell^{-1/2}\theta_\ell\right)^\top\left(\rho_\ell - \theta_\ell^\top q_\ell^{-1}\theta_\ell\right)^{-1}
\end{cases}
\tag{94}
$$

where the proximals $\mathrm{prox}_g$ and $\mathrm{prox}_\ell$ respectively refer to the arginf in $Q$ (resp. $x_\ell$) of the envelopes $\mathcal{M}_g$ (89) (resp. $\mathcal{M}$ 91).

## C.7 Saddle-point equations : $\ell_2$

We now specialize the saddle-point equations (93) to the case of an $\ell_2$ regularizer $g(\cdot) = 1/2\|\cdot\|$ the entropic potential admits the simple form

$$
\begin{aligned}
\Psi_Q &= \frac{1}{2d}\mathrm{Tr}\left[\left(\lambda\mathbb{I}_r \odot \mathbb{I}_d + \sum_{\ell=1}^{L}\hat{V}_\ell \otimes \boldsymbol{\Sigma}_\ell\right)^{-1} \odot \left(\sum_{\ell=1}^{L}\hat{q}_\ell \otimes \boldsymbol{\Sigma}_\ell + \left(\sum_{\ell=1}^{L}(\sqrt{d}\hat{m}_\ell p_\ell^\top + \hat{\theta}_\ell Q_\star^\top \boldsymbol{\Sigma}_\ell)\right)^{\otimes 2}\right)\right] \\
&= \frac{1}{2d}\sum_{i=1}^{d}\mathrm{Tr}\left[\left(\lambda + \sum_{\ell=1}^{L}\lambda_i^\ell\hat{V}_\ell\right)^{-1}\left(\sum_{\ell=1}^{L}\lambda_i^\ell\hat{q}_k + \left(\sum_{\ell=1}^{L}(\sqrt{d}\hat{m}_\ell p_\ell^\top e_i + \hat{\theta}_\ell Q_\star^\top \Sigma_\ell e_i)\right)\left(\sum_{\ell=1}^{L}(\sqrt{d}\hat{m}_\ell p_\ell^\top e_i + \hat{\theta}_\ell Q_\star^\top \Sigma_\ell e_i)\right)^\top\right)\right] \\
&\overset{d\to\infty}{=} \frac{1}{2}\int d\nu(\gamma,\tau,\pi)\mathrm{Tr}\left[\left(\lambda + \sum_{\ell=1}^{L}\gamma_\ell\hat{V}_\ell\right)^{-1}\left(\sum_{\ell=1}^{L}\gamma_\ell\hat{q}_\ell + \left(\sum_{\ell=1}^{L}\tau_\ell\hat{m}_\ell + \gamma_\ell\hat{\theta}_\ell \cdot \pi\right)^{\otimes 2}\right)\right].
\end{aligned}
\tag{95}
$$

The replica free energy thus reads

$$
\begin{aligned}
\Phi &= \frac{1}{2}\sum_{\ell=1}^{L}\left(\mathrm{Tr}\,\hat{V}_\ell q_\ell - \mathrm{Tr}\,\hat{q}_\ell V_\ell\right) - \sum_{\ell=1}^{L}\hat{m}_\ell^\top m_\ell - \sum_{\ell=1}^{L}\mathrm{Tr}\,\hat{\theta}_\ell^\top\theta_\ell - \alpha\mathbb{E}_{y,\xi}\mathcal{M}(y,\xi) \\
&\quad + \frac{1}{2}\int d\nu(\gamma,\tau,\pi)\mathrm{Tr}\left[\left(\lambda + \sum_{\ell=1}^{L}\gamma_\ell\hat{V}_\ell\right)^{-1}\left(\sum_{\ell=1}^{L}\gamma_\ell\hat{q}_\ell + \left(\sum_{\ell=1}^{L}\tau_\ell\hat{m}_\ell + \gamma_\ell\hat{\theta}_\ell \cdot \pi\right)^{\otimes 2}\right)\right],
\end{aligned}
\tag{96}
$$

leading to the saddle point equations

$$
\begin{cases}
\hat{q}_\ell = \alpha\mathbb{E}_{\Xi,Y}V_\ell^{-1}\left(\mathrm{prox}_\ell - q_\ell^{\frac{1}{2}}\xi_\ell - m_\ell\right)^{\otimes 2}V_\ell^{-1} \\
\hat{V}_\ell = \hat{\theta}_\ell\theta_\ell^\top q_\ell^{-1} - \alpha\mathbb{E}_{\Xi,Y}V_\ell^{-1}\left(\mathrm{prox}_\ell - q_\ell^{\frac{1}{2}}\xi_\ell - m_\ell\right)\xi_\ell^\top q_\ell^{-\frac{1}{2}} \\
\hat{m}_\ell = \alpha\mathbb{E}_{\xi,\eta}V_\ell^{-1}\left(\mathrm{prox}_\ell - q_\ell^{\frac{1}{2}}\xi_\ell - m_\ell\right) \\
\hat{\theta}_\ell = \alpha\mathbb{E}_{\xi,\eta}V_\ell^{-1}\left(\mathrm{prox}_\ell - q_\ell^{\frac{1}{2}}\xi_\ell - m_\ell\right)\left(y_\ell - \xi_\ell^\top q_\ell^{-1/2}\theta_\ell\right)^\top\left(\rho_\ell - \theta_\ell^\top q_\ell^{-1}\theta_\ell\right)^{-1}
\end{cases}
$$

$$
\begin{cases}
q_\ell = \int d\nu(\gamma,\tau,\pi)\gamma_\ell\left(\lambda\mathbb{I}_r + \sum_{\kappa=1}^L\gamma_\kappa\hat{V}_\kappa\right)^{-1}\left(\sum_{\kappa=1}^L\gamma_\kappa\hat{q}_\kappa + \left(\sum_{\kappa=1}^L\hat{m}_\kappa\tau_\kappa + \gamma_\kappa\hat{\theta}_\kappa\cdot\pi\right)^{\otimes 2}\right)\left(\lambda\mathbb{I}_r + \sum_{\kappa=1}^L\gamma_\kappa\hat{V}_\kappa\right)^{-1} \\
V_\ell = \int d\nu(\gamma,\tau,\pi)\gamma_\ell\left(\lambda\mathbb{I}_r + \sum_{\kappa=1}^L\gamma_\kappa\hat{V}_\kappa\right)^{-1} \\
m_\ell = \int d\nu(\gamma,\tau,\pi)\tau_\ell\left(\lambda\mathbb{I}_r + \sum_{\kappa=1}^L\gamma_\kappa\hat{V}_\kappa\right)^{-1}\left(\sum_{\kappa=1}^L\hat{m}_\kappa\tau_\kappa + \gamma_\kappa\hat{\theta}_\kappa\cdot\pi\right) \\
\theta_\ell = \int d\nu(\gamma,\tau,\pi)\gamma_\ell\left(\lambda\mathbb{I}_r + \sum_{\kappa=1}^L\gamma_\kappa\hat{V}_\kappa\right)^{-1}\left(\sum_{\kappa=1}^L\hat{m}_\kappa\tau_\kappa + \gamma_\kappa\hat{\theta}_\kappa\cdot\pi\right)\pi^\top.
\end{cases}
$$

$$(97)$$

which finishes to recover (7). Let us finally mention that the update equations (7) for the summary statistics (6) do *not* describe the dynamics of gradient descent, but rather that of an Approximate Message Passing algorithm [63], which we elicit in Appendix B for completeness. $\qquad\square$

## C.8  test MSE

The generalization performance is measured by the test error

$$
\epsilon_g \equiv \frac{1}{L}\mathbb{E}_{\mathcal{D}}\mathbb{E}_x\left\|\mathtt{T}\left[\frac{1}{\sqrt{d}}xQ_\star\right]x - \mathtt{S}\left[\frac{1}{\sqrt{d}}(x+p)\hat{Q}\right](x+p)\right\|^2. \tag{98}
$$

Expliciting this expression in terms of the correlated Gaussian variables $xQ_\star, xQ$ allows to straightforwardly show that $\epsilon_g$ admits the sharp asymptotic characterization in terms of the summary statistics characterized by (97):

$$
\epsilon_g = \frac{1}{L}\mathbb{E}_X\,\mathrm{Tr}\left[\mathtt{S}[X]\rho_\Sigma\mathtt{S}[X]^\top\right] + \frac{1}{L}\mathbb{E}_Y\,\mathrm{Tr}\left[\mathtt{T}[Y]\rho_\Sigma\mathtt{T}[Y]^\top\right] - 2\frac{1}{L}\mathbb{E}_{X,Y}\,\mathrm{Tr}\left[\mathtt{S}[X]\rho_\Sigma\mathtt{T}[Y]^\top\right], \tag{99}
$$

where the average bears on $X \in \mathbb{R}^{L\times r}, Y \in \mathbb{R}^{L\times t}$ with independent rows with statistics

$$
(x_\ell, y_\ell) \sim \mathcal{N}\left[\begin{pmatrix}m \\ 0\end{pmatrix}, \left(\begin{array}{c|c}q_\ell & \theta_\ell \\ \hline \theta_\ell^\top & \rho_\ell\end{array}\right)\right] \tag{100}
$$

## C.9  Training loss

We finally turn to the training loss. It is reasonable to expect, from statistical physics, that the training loss should be equal to the free energy $-\Phi$ at zero temperature. We provide below an alternative derivation, for simplicity in the case of $\ell_2$ regularization $g = 1/2\|\cdot\|^2$. First note that the training loss $\epsilon_t$ can be expressed as

$$
\epsilon_t = -\lim_{\beta\to\infty}\partial_\beta\underbrace{\frac{1}{d}\ln Z(\beta)}_{\Phi(\beta)} \tag{101}
$$

Where $\Phi(\beta)$ is the free entropy at finite temperature. The trace potential $\Psi_t$ bears no explicit dependence on $\beta$. On the other hand,

$$
\beta\Psi_Q = -\frac{1}{2}\ln\det\left[\beta\lambda\mathbb{I}_r\otimes\mathbb{I}_d + \sum_{\ell=1}^L\hat{V}_\ell\otimes\Sigma_\ell\right]
$$

$$
+ \frac{1}{2d}\mathrm{Tr}\left[\left(\beta\lambda\mathbb{I}_r\odot\mathbb{I}_d + \sum_{\ell=1}^L\hat{V}_\ell\otimes\boldsymbol{\Sigma}_\ell\right)^{-1}\odot\left(\sum_{\ell=1}^L\hat{q}_\ell\otimes\boldsymbol{\Sigma}_\ell + \left(\sum_{\ell=1}^L(\sqrt{d}\hat{m}_\ell p_\ell^\top + \hat{\theta}_\ell Q_\star^\top\Sigma_\ell\right)^{\otimes 2}\right)\right]
$$

$$(102)$$

Thus

$$\partial_\beta(\beta\Psi_Q) = -\frac{\lambda}{2}\,\mathrm{Tr}\left[\beta\lambda\mathbb{I}_r\otimes\mathbb{I}_d + \sum_{\ell=1}^{L}\hat{V}_\ell\otimes\Sigma_\ell\right]^{-1}$$
$$-\frac{\lambda}{2d}\,\mathrm{Tr}\left[\left(\beta\lambda\mathbb{I}_r\odot\mathbb{I}_d + \sum_{\ell=1}^{L}\hat{V}_\ell\otimes\boldsymbol{\Sigma}_\ell\right)^{-2}\odot\left(\sum_{\ell=1}^{L}\hat{q}_\ell\otimes\boldsymbol{\Sigma}_\ell + \left(\sum_{\ell=1}^{L}(\sqrt{d}\hat{m}_\ell p_\ell^\top + \hat{\theta}_\ell Q_\star^\top\Sigma_\ell)\right)^{\otimes2}\right)\right]$$

(103)

Finally, going through the same rescaling steps to take the $\beta\to\infty$ limit,

$$\lim_{\beta\to\infty}\partial_\beta(\beta\Psi_Q) = -\frac{\lambda}{2d}\,\mathrm{Tr}\left[\left(\lambda\mathbb{I}_r\odot\mathbb{I}_d + \sum_{\ell=1}^{L}\hat{V}_\ell\otimes\boldsymbol{\Sigma}_\ell\right)^{-2}\odot\left(\sum_{\ell=1}^{L}\hat{q}_\ell\otimes\boldsymbol{\Sigma}_\ell + \left(\sum_{\ell=1}^{L}(\sqrt{d}\hat{m}_\ell p_\ell^\top + \hat{\theta}_\ell Q_\star^\top\Sigma_\ell)\right)^{\otimes2}\right)\right]$$

(104)

By the same token, it is straightforward to see that

$$\lim_{\beta\to\infty}\partial_\beta(\beta\Psi_y) = -\mathbb{E}_{Y,\Xi}\left[\mathcal{M}(Y,\Xi) - \frac{1}{2}\sum_{\ell=1}^{L}\mathrm{Tr}\left[\underbrace{V_\ell^{-1}\left(x_\ell - q_\ell^{1/2}\xi_\ell - m_\ell\right)^{\otimes2}}_{\hat{q}_\ell V_\ell}\right]\right]$$

(105)

We used the self-consistent equations (7) to identify the term in underbrace. Putting everything together,

$$-\epsilon_t = \lim_{\beta\to\infty}\partial_\beta\Psi(\beta) = -\frac{\lambda}{2}\int d\nu(\gamma,\tau)\,\mathrm{Tr}\left[\left(\lambda + \sum_{\ell=1}^{L}\gamma_\ell\hat{V}_\ell\right)^{-1}\left(\sum_{\ell=1}^{L}\gamma_\ell\hat{q}_\ell + \left(\sum_{\ell=1}^{L}\tau_\ell\hat{m}_\ell + \hat{\theta}_\ell\cdot\pi\right)^{\otimes2}\right)\right]$$
$$-\alpha\mathbb{E}_{Y,\Xi}\left[\mathcal{M}(Y,\Xi)\right] + \frac{1}{2}\sum_{\ell=1}^{L}\mathrm{Tr}[\hat{q}_\ell V_\ell].$$

(106)

This constitutes a sharp asymptotic characterization of the training loss $\epsilon_t$ as a function of the summary statistics characterized in Result 4.2.

For completeness, we finally explicit the connection between $\epsilon_t$ and the negative free entropy (i.e. the *free energy* in statistical physics). We go back to massage the expression for the free entropy

$$\Phi = \frac{1}{2}\sum_{\ell=1}^{L}\left(\mathrm{Tr}\,\hat{V}_\ell q_\ell - \mathrm{Tr}\,\hat{q}_\ell V_\ell\right) - \sum_{\ell=1}^{L}\hat{m}_\ell^\top m_\ell - \sum_{\ell=1}^{L}\mathrm{Tr}\,\hat{\theta}_\ell^\top\theta_\ell - \alpha\mathbb{E}_{Y,\Xi}\mathcal{M}(Y,\Xi)$$
$$+\frac{1}{2}\int d\nu(\gamma,\tau)\,\mathrm{Tr}\left[\left(\lambda + \sum_{\ell=1}^{L}\gamma_\ell\hat{V}_\ell\right)^{-1}\left(\sum_{\ell=1}^{L}\gamma_\ell\hat{q}_\ell + \left(\sum_{\ell=1}^{L}\tau_\ell\hat{m}_\ell + \gamma_\ell\hat{\theta}_\ell\cdot\pi\right)^{\otimes2}\right)\right]$$
$$=\frac{1}{2}\sum_{\ell=1}^{L}\mathrm{Tr}\left[\hat{q}_\ell V_\ell + \hat{V}_\ell q_\ell\right] - \frac{1}{2}\int d\nu(\gamma,\tau)\,\mathrm{Tr}\left[\left(\lambda + \sum_{\ell=1}^{L}\gamma_\ell\hat{V}_\ell\right)^{-1}\left(\sum_{\ell=1}^{L}\gamma_\ell\hat{q}_\ell + \left(\sum_{\ell=1}^{L}\tau_\ell\hat{m}_\ell + \gamma_\ell\hat{\theta}_\ell\cdot\pi\right)^{\otimes2}\right)\right]$$
$$-\alpha\mathbb{E}_{Y,\Xi}\mathcal{M}(Y,\Xi)$$
$$=\frac{1}{2}\sum_{\ell=1}^{L}\mathrm{Tr}[\hat{q}_\ell V_\ell] + \frac{1}{2}\int d\nu(\gamma,\tau)\,\mathrm{Tr}\left[\left(\lambda + \sum_{\ell=1}^{L}\gamma_\ell\hat{V}_\ell\right)^{-2}\left(\sum_{\ell=1}^{L}\hat{V}_\ell\gamma_\ell - \lambda - \sum_{\ell=1}^{L}\gamma_\ell\hat{V}_\ell\right)\left(\sum_{\ell=1}^{L}\gamma_\ell\hat{q}_\ell + \left(\sum_{\ell=1}^{L}\tau_\ell\hat{m}_\ell + \gamma_\ell\hat{\theta}_\ell\cdot\pi\right)^{\otimes2}\right)\right]$$
$$-\alpha\mathbb{E}_{Y,\Xi}\mathcal{M}(Y,\Xi)$$
$$=-\frac{\lambda}{2}\int d\nu(\gamma,\tau)\,\mathrm{Tr}\left[\left(\lambda + \sum_{\ell=1}^{L}\gamma_\ell\hat{V}_\ell\right)^{-1}\left(\sum_{\ell=1}^{L}\gamma_\ell\hat{q}_\ell + \left(\sum_{\ell=1}^{L}\tau_\ell\hat{m}_\ell + \hat{\theta}_\ell\cdot\pi\right)^{\otimes2}\right)\right]$$
$$-\alpha\mathbb{E}_{Y,\Xi}\left[\mathcal{M}(Y,\Xi)\right] + \frac{1}{2}\sum_{\ell=1}^{L}\mathrm{Tr}[\hat{q}_\ell V_\ell]$$
$$=-\epsilon_t$$

(107)

In other words, the training loss is equal to the zero-temperature free energy.

## C.10  Extensions

For the sake of clarity and definiteness, we stated the technical results for in the simplest instance, under notably the symplifying assumptions of independent tokens, tied key and query weight matrices, as exposed in Section 3 of the main text. These two assumptions can in fact be relaxed. In this final subsection of Appendix C, we discuss how the characterization of Result 4.2 can be generalized to more complex data distributions and attention architectures.

**Data model**  Let us consider a data distribution allowing for statistical correlations between different tokens. A simple model meeting this criterion, proposed in [73], consists in assuming every token (row) $x_\ell$ is drawn from a Gaussian mixture with $K_\ell$ with respective probabilities $\rho_{\ell,k}$,

$$\boldsymbol{x}_\ell, c_\ell \sim \sum_{k=1}^{K_\ell} \rho_{\ell,k} \delta_{c_\ell,k} \mathcal{N}(\mu_{\ell,k}, \Sigma_{\ell,k}). \tag{108}$$

In (108), the random variable $c_\ell \in [K_\ell]$ indicates the index of the cluster the $\ell-$th token is drawn from. Note that we generically allow for different rows to be sampled from distinct mixtures. In order to introduce correlations between different tokens, we further assume that the variables $c = (c_1, ..., c_L)$ follow a generic joint law $\rho(c)$. Intuitively, the different models can model different types of words (e.g. verbs, pronouns, adjectives), and the nature of the words, as reflected by the variable $c_\ell$, is correlated across tokens. We assume that $\|\mu_{\ell,k}\| = \Theta_d(1)$, i.e. are of the same order as the positional encodings.

**Assumption C.1.** *Finally, similarly to Assumption 4.1, let us assume that the set of matrices $\{\{\Sigma_{\ell,k}\}_{k=1}^{K_\ell}\}_{\ell=1}^L$ admits a common set of eingevectors $\{e_i\}_{i=1}^d$ with eigenvalues $\{\lambda_i^{\ell,k}\}_{i=1}^d$. The eigenvalues $\{\lambda_i^{\ell,k}\}_{\ell,k,i}$ and the projection of the cluster means (to which the positional encodings are added) $\{\mu_{\ell,k} + p_\ell\}_{\ell,k}$ and the teacher columns $\{Q_i^\star\}_{i=1}^{r_t}$ on these eigenvectors are assumed to admit a well-defined joint distribution $\nu$ as $d \to \infty$ – namely, for $\gamma = (\gamma_{\ell,k})_{\ell,k}$, $\pi = (\pi_1, ..., \pi_t) \in \mathbb{R}^{r_t}$, $\tau = (\tau_{\ell,k})_{\ell,k}$:*

$$\frac{1}{d} \sum_{i=1}^d \prod_{\ell=1}^L \prod_{k=1}^{K_\ell} \delta\left(\lambda_i^{\ell,k} - \gamma_{\ell,k}\right) \delta\left(\sqrt{d} e_i^\top (\mu_{\ell,k} + p_\ell) - \tau_{\ell,k}\right) \prod_{j=1}^{r_t} \delta\left(e_i^\top Q_j^\star - \pi_j\right) \xrightarrow{d\to\infty} \nu\left(\gamma, \tau, \pi\right). \tag{109}$$

**Architecture**  Again in the interest of further generality, we alleviate the architectural restrictions on the attention (2), and allow for *untied* key and query weight matrices. More precisely, let us consider function of the form

$$f_{Q,K}(x) = \mathbf{S}\left[\frac{1}{\sqrt{d}}(x+p)Q, \frac{1}{\sqrt{d}}(x+p)K\right](x+p), \tag{110}$$

parameterized by two set of weights $Q, K \in \mathbb{R}^{d \times r_s}$. In order to work with more compact expression, let us introduce the total weight matrix $W \in \mathbb{R}^{d \times (r_s + r_s)}$, obtained as the horizontal concatenation of $Q, K$. Then (110) can be written compactly as

$$f_{K,Q}(x) = g_W(x) \equiv \mathbf{G}\left[\frac{1}{\sqrt{d}}(x+p)W\right](x+p), \tag{111}$$

where for any argument $h \in \mathbb{R}^{L \times (r_s + r_s)}$, wiewed as a $1 \times 2$ block matrix with two blocks $h_1, h_2 \in \mathbb{R}^{L \times r}$, $\mathbf{G}(h) = \mathbf{S}(h_1, h_2)$.

**Asymptotic characterization**  A sharp asymptotic characterization of the test error and train loss for this more general model can be derived along the same lines, as presented in Appendix C (see also [73]). For the sake of conciseness, we only report the final result.

**Result C.2.** *Under Assumption C.1, in the limit $n, d \to \infty$, $\|\boldsymbol{p}\|$, $n/d, L, r_s, r_t = \Theta_d(1)$, the summary statistics*

$$\rho_{\ell,k} \equiv \frac{Q_\star^\top \boldsymbol{\Sigma}_{\ell,k} Q_\star}{d} \in \mathbb{R}^{r_t \times r_t}, \qquad\qquad q_{\ell,k} \equiv \frac{\hat{W}^\top \boldsymbol{\Sigma}_{\ell,k} \hat{W}}{d} \in \mathbb{R}^{2r_s \times 2r_s},$$

$$m_{\ell,k} \equiv \frac{\hat{W}^\top (\mu_{\ell,k} + p_\ell)}{d} \in \mathbb{R}^{2r_s}, \qquad\qquad \theta_{\ell,k} \equiv \frac{\hat{W}^\top \boldsymbol{\Sigma}_{\ell,k} Q_\star}{d} \in \mathbb{R}^{2r_s \times r_t} \qquad (112)$$

*concentrate in probability, and are solutions of the set of finite-dimensional self-consistent equations*

$$\begin{cases} \hat{q}_{\ell,k} = \alpha \mathbb{E}_c \delta_{c_\ell,k} \mathbb{E}_{\Xi,Y} V_{\ell,k}^{-1} \left( \text{prox}_\ell^c - q_{\ell,k}^{\frac{1}{2}} \xi_\ell - m_{\ell,k} \right)^{\otimes 2} V_{\ell,k}^{-1} \\ \hat{V}_{\ell,k} = \hat{\theta}_{\ell,k} \theta_{\ell,k}^\top q_{\ell,k}^{-1} - \alpha \mathbb{E}_c \delta_{c_\ell,k} \mathbb{E}_{\Xi,Y} V_{\ell,k}^{-1} \left( \text{prox}_\ell^c - q_{\ell,k}^{\frac{1}{2}} \xi_\ell - m_{\ell,k} \right) \xi_\ell^\top q_{\ell,k}^{-\frac{1}{2}} \\ \hat{m}_{\ell,k} = \alpha \mathbb{E}_c \delta_{c_\ell,k} \mathbb{E}_{\Xi,Y} V_{\ell,k}^{-1} \left( \text{prox}_\ell^c - q_{\ell,k}^{\frac{1}{2}} \xi_\ell - m_{\ell,k} \right) \\ \hat{\theta}_{\ell,k} = \alpha \mathbb{E}_c \delta_{c_\ell,k} \mathbb{E}_{\Xi,Y} V_{\ell,k}^{-1} \left( \text{prox}_\ell^c - q_{\ell,k}^{\frac{1}{2}} \xi_\ell - m_{\ell,k} \right) \left( y_\ell - \theta_{\ell,k}^\top q_{\ell,k}^{-1/2} \xi_\ell \right)^\top \left( \rho_{\ell,k} - \theta_{\ell,k}^\top q_{\ell,k}^{-1} \theta_{\ell,k} \right)^{-1} \end{cases}$$

$$(113)$$

$$\begin{cases} q_{\ell,k} = \int d\nu(\gamma, \tau, \pi) \gamma_{\ell,k} \left( \lambda \mathbb{I}_{2r_s} + \sum_\kappa \sum_j \gamma_{\kappa,j} \hat{V}_{\kappa,j} \right)^{-1} \\ \qquad \left[ \left( \sum_\kappa \sum_j \hat{m}_{\kappa,j} \tau_{\kappa,j} + \gamma_{\kappa,j} \hat{\theta}_{\kappa,j} \pi \right)^{\otimes 2} + \sum_\kappa \sum_j \gamma_{\kappa,j} \hat{q}_{\kappa,j} \right] \left( \lambda \mathbb{I}_{2r_s} + \sum_\kappa \sum_j \gamma_{\kappa,j} \hat{V}_{\kappa,k} \right)^{-1} \\ V_{\ell,k} = \int d\nu(\gamma, \tau, \pi) \gamma_{\ell,k} \left( \lambda \mathbb{I}_{2r_s} + \sum_\kappa \sum_j \gamma_{\kappa,j} \hat{V}_{\kappa,j} \right)^{-1} \\ m_{\ell,k} = \int d\nu(\gamma, \tau, \pi) \tau_{\ell,k} \left( \lambda \mathbb{I}_{2r_s} + \sum_\kappa \sum_j \gamma_{\kappa,j} \hat{V}_{\kappa,j} \right)^{-1} \left( \sum_\kappa \sum_j \hat{m}_{\kappa,j} \tau_{\kappa,j} + \gamma_{\kappa,j} \hat{\theta}_{\kappa,j} \pi \right) \\ \theta_{\ell,k} = \int d\nu(\gamma, \tau, \pi) \gamma_{\ell,k} \left( \lambda \mathbb{I}_{2r_s} + \sum_\kappa \sum_j \gamma_{\kappa,j} \hat{V}_{\kappa,j} \right)^{-1} \left( \sum_\kappa \sum_j \hat{m}_{\kappa,j} \tau_{\kappa,j} + \gamma_{\kappa,j} \hat{\theta}_{\kappa,j} \pi \right) \pi^\top \end{cases}$$

$$(114)$$

*The averages bears over $\Xi \in \mathbb{R}^{L \times 2r_s}, Y \in \mathbb{R}^{L \times r_t}$, with rows $\xi_\ell \sim \mathcal{N}(0, \mathbb{I}_{2r_s})$ independently and $y_\ell \sim \mathcal{N}(\xi_\ell^\top q_{\ell,c_\ell}^{-1/2} \theta_{\ell,c_\ell} \rho_{\ell,c_\ell} - \theta_{\ell,c_\ell}^\top q_{\ell,c_\ell}^{-1} \theta_{\ell,c_\ell})$, conditionally on c. $\text{prox}_\ell^c$ corresponds to the arginf of the minimization defining the Moreau envelope $\mathcal{M}(c, Y, \Xi)$, as*

$$\text{prox}^c = \underset{X}{\text{arginf}} \left\{ \frac{1}{2} \sum_{\ell=1}^L \text{Tr} \left[ V_{\ell,c_\ell}^{-1} \left( X_\ell - q_{\ell,c_\ell}^{1/2} \xi_\ell - m_{\ell,c_\ell} \right)^{\otimes 2} \right] + \ell \left( Y + m_\star^c, X, c \right) \right\}. \qquad (115)$$

*We introduced $m_\star^c \in \mathbb{R}^{L \times r_t}$, whose rows are*

$$(m_\star^c)_\ell = \frac{\mu_{\ell,c_\ell}^\top Q_\star}{\sqrt{d}}, \qquad (116)$$

*and the shorthand*

$$\ell(y, x, c) = \text{Tr} \left[ \texttt{G}[x] \rho_\Sigma^c \texttt{G}[x]^\top \right] - 2 \text{Tr} \left[ \texttt{T}[y] \rho_\Sigma^c \texttt{G}[x]^\top \right], \qquad (117)$$

*where $\rho_\Sigma^c \in \mathbb{R}^{L \times L}$ is diagonal with elements*

$$(\rho_\Sigma^c)_{\ell\ell} = \int d\nu(\gamma, \tau, \pi) \gamma_{\ell,c_\ell}. \qquad (118)$$

**Test error**   *The test error admits the sharp asymptotic characterization*

$$\epsilon_g = \frac{1}{L}\mathbb{E}_c\mathbb{E}_{h,h^\star}\left[\mathrm{Tr}\big[\mathtt{T}[h^\star]\rho_\Sigma^c\mathtt{T}[h^\star]^\top\big] + \mathrm{Tr}\big[\mathtt{G}[h]\rho_\Sigma^c\mathtt{G}[h]^\top\big] - 2\,\mathrm{Tr}\big[\mathtt{T}[h^\star]\rho_\Sigma^c\mathtt{G}[h]^\top\big]\right], \qquad (119)$$

*where, conditioned on the class assignments c, the average bears on $h \in \mathbb{R}^{L\times 2r_s}$, $Y \in \mathbb{R}^{L\times r_t}$ with independent rows with statistics*

$$(h_\ell, h_\ell^\star) \sim \mathcal{N}\left[\begin{pmatrix} m_{\ell,c_\ell} \\ m_{\ell,c_\ell}^\star \end{pmatrix}, \left(\begin{array}{c|c} q_{\ell,c_\ell} & \theta_{\ell,c_\ell} \\ \hline \theta_{\ell,c_\ell}^\top & \rho_{\ell,c_\ell} \end{array}\right)\right]. \qquad (120)$$

*Finally, the train loss $\epsilon_t$ converges in probability to*

$$\epsilon_t = \frac{\lambda}{2}\int d\nu(\gamma,\tau,\pi)\,\mathrm{Tr}\left[\left(\lambda\mathbb{I}_{2r_s} + \hat{v} + \sum_\kappa\sum_k \gamma_{\kappa,k}\hat{V}_{\kappa,k}\right)^{-2}\left[\left(\underset{\kappa}{\textstyle\sum}\underset{k}{\textstyle\sum}\hat{m}_{\kappa,k}\tau_{\kappa,k} + \gamma_{\kappa,k}\hat{\theta}_{\kappa,k}\pi\right)^{\otimes 2} + \underset{\kappa}{\textstyle\sum}\underset{k}{\textstyle\sum}\gamma_{\kappa,k}\hat{q}_{\kappa,k}\right]\right]$$

$$+ \alpha\mathbb{E}_{c,Y,\Xi}\mathcal{M}(c,Y,\Xi) - \frac{1}{2}\sum_{\ell=1}^{L}\sum_{k=1}^{K_\ell}\mathrm{Tr}[\hat{q}_{\ell,k}V_{\ell,k}]. \qquad (121)$$

**Trainable value matrix**   Finally, let us remark that it should be possible to leverage a similar derivation to further accommodate trainable value matrices $V$, provided they are low rank, i.e. in $\mathbb{R}^{d\times r_V}$ with $r_V = \Theta_d(1)$ as $d \to \infty$. Similarly, introducing the stacked weights $W = (Q|K|V) \in \mathbb{R}^{d\times(2r_s+r_V)}$ allows to perform the asymptotic analysis. Since this architectural modification then implies that $f$ now takes values in $\mathbb{R}^{L\times r_V}$, one cannot immediately analyze this architecture for the task considered in the present manuscript, as the labels are valued in $\mathbb{R}^{L\times d}$, and the setting and questions explored would need to be entirely modified. We thus postpone a thorough analysis of the effect of a trained value matrix to future investigations.

## D   Derivation of Result 5.1

In this Appendix we derive the asymptotic characterization of the learning performance of the dense linear baseline (15), as stated in Result 5.1. Consider the empirical risk minimization (16)

$$\mathcal{R}(W) = \frac{1}{n}\sum_{\mu=1}^{n}\|A^\star(x^\mu)x^\mu - Wx^\mu\|^2 \qquad (122)$$

where we use the shorthand notation $A^\star(x) \equiv \mathtt{T}[1/\sqrt{d}xQ_\star]$ for the target attention score matrix (1). The expression for the risk can be asymptotically simplified as

$$\mathcal{R}(W) = \frac{1}{n}\sum_{\mu=1}^{n}\left[A^\star(x^\mu)\rho_\Sigma A^\star(x^\mu)^\top - (A^\star(x^\mu)\rho_\Sigma W^\top + \mathrm{h.c}) + W\rho_\Sigma W^\top + o(1/\sqrt{d})\right]$$

$$\approx \mathbb{E}_x\left[A^\star(x)\rho_\Sigma A^\star(x)^\top\right] - (W\rho_\Sigma\mathbb{E}_x\left[A^\star(x)\right]^\top + \mathrm{h.c}) + W\rho_\Sigma W^\top$$

$$= \mathbb{E}_x\left[A^\star(x)\rho_\Sigma A^\star(x)^\top\right] - \mathbb{E}_x\left[A^\star(x)\right]\rho_\Sigma\mathbb{E}_x\left[A^\star(x)\right]^\top + \left\|\rho_\Sigma^{1/2}\left(W - \mathbb{E}_x\left[A^\star(x)\right]\right)\right\|^2 \qquad (123)$$

Therefore the learnt weight $\hat{W}$ is simply equal to

$$\hat{W} = \mathbb{E}_x\left[A^\star(x)\right] = \mathbb{E}_x\left[\mathtt{T}[1/\sqrt{d}xQ_\star]\right]. \qquad (124)$$

Naming $h = 1/\sqrt{d}xQ_\star$ the argument of the last term, the matrix $h$ possesses independent rows with statistics

$$h_\ell \sim \mathcal{N}(0, \rho_\ell) \qquad (125)$$

where we remind that $\rho_\ell \equiv 1/dQ_\star^\top\Sigma_\ell Q_\star$. Therefore, the trained weights $\hat{W}$ obtained by minimizing the *empirical* loss also coincide with the minimizer of the *population* loss. Intuitively, this follows from the fact that in the asymptotic limit considered $n, d \to \infty$ and $L = \Theta_d(1)$, training $W$, i.e. a number $L^2 = \Theta_d(1)$ of parameters on the empirical loss for $n \ll L^2$ data points is equivalent

asymptotically to directly training on the population loss. Expliciting the corresponding test MSE, one reaches the sharp asymptotic characterization

$$\epsilon_g^{\text{lin}} = \frac{1}{L} \text{Tr}\left[\hat{W}\rho_\Sigma \hat{W}^\top\right] + \frac{1}{L}\mathbb{E}_h \text{Tr}\left[\mathtt{T}[h]\rho_\Sigma \mathtt{T}[h]^\top\right]$$
$$- \frac{2}{L}\mathbb{E}_h \text{Tr}\left[\hat{W}\rho_\Sigma \mathtt{T}[h]^\top\right]. \tag{126}$$

We close this Appendix by giving a few examples for definiteness.

### D.1 Examples

**Purely positional target**  Let us consider the case where the target is purely positional, i.e. $\mathtt{T}[x] = A$ for all $x$. This corresponds to the $\omega = 1$ limit of the target (14). It then follows from Result 5.1 that the dense linear layer recover perfectly the target weights $\hat{W} = A$.

**Target** (14)  For the target discussed in the main text (14),

$$\hat{W} = (1 - \omega)\mathbb{E}_h \text{softmax}(hh^\top) + \omega A \tag{127}$$

with $h$ and having independent rows $h_\ell \sim \mathcal{N}(0, \rho_\ell)$.

## E  Supplementary Experiments

### E.1  Empirical scaling of $\alpha = d/n$

In the following we verify that our experiments are consistent with the scaling behaviour predicted from the theory. We jointly increase $d$ and $n$ for a fixed value of $\alpha$. In Fig. 7 we indeed observe the expected behaviour for an exemplary value of $\alpha = 2$. The same holds for the summary statistics $\theta$ and $m$, which concentrate as $d$ and $n$ jointly grow, shown in Fig. 3 (left) in the main text.

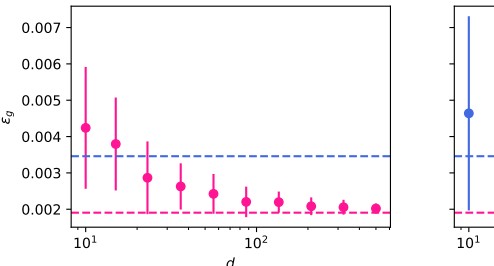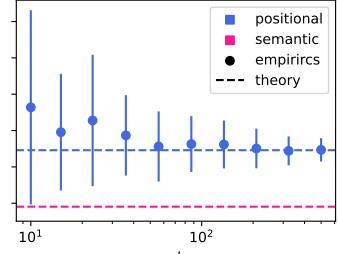

Figure 7: Scaling $d$ and $n$ jointly for $\alpha = 1.5$ approaches the theoretical prediction of the generalization error of the positional and semantic local minima. Experimental settings as in Fig. 2, with 70 runs per datapoint.

### E.2  Alternative hyperparameters

We provide supplementary results for different parameter settings. Fig. 8 on the left shows more slices from the phase diagram that appears in the main Fig. 3. For the experimental section of the main text, we chose a specific $A$ for definiteness. In the following, we present the same results for a different $A$ with a stronger off-diagonal and a higher rank,

$$A = \begin{pmatrix} 0.3 & 0.7 \\ 0.8 & 0.2 \end{pmatrix}. \tag{128}$$

In Fig. 9 we present the analogous simulations to Fig. 3 (center, right). While the global phenomena match the previous example, the details of the transitions location differ.

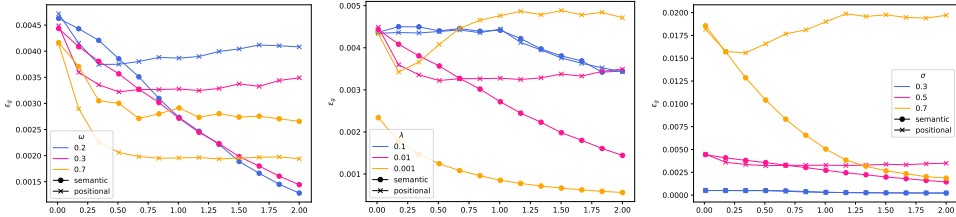

Figure 8: *Alternative Parameters.* Mixed positional/semantic teacher for $\omega = 0.3$. Settings is $r_s = r_t = 1, L = 2, A = ((0.6, 0.4), (0.4, 0.6)), \boldsymbol{\Sigma}_1 = \boldsymbol{\Sigma}_2 = 0.25\mathbb{I}_d, \boldsymbol{p}_1 = \boldsymbol{1}_d = -\boldsymbol{p}_2$ and $\boldsymbol{Q}_\star \sim \mathcal{N}(0, \mathbb{I}_d)$. While keeping all other settings the same, we vary from left to right: The target positionality $\omega$, the student regularizer $\lambda$ and the standard deviation $\sigma$ (which is $0.5 = \sqrt{\boldsymbol{\Sigma}_1}$) . Experiment settings as in Fig. 2.

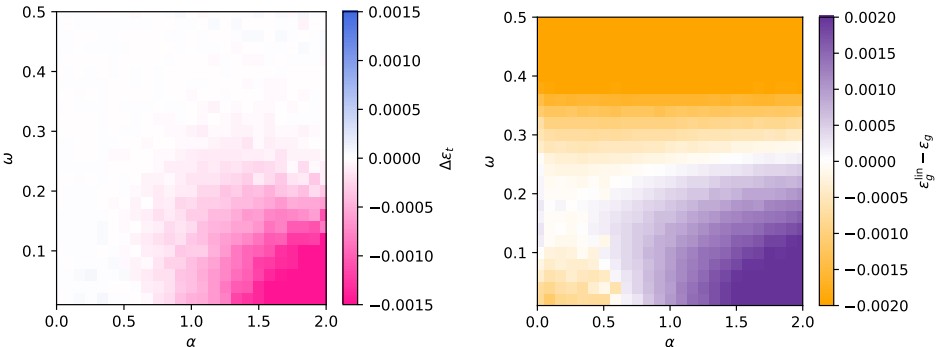

Figure 9: *Alternative Positional Matrix.* $r_s = r_t = 1, L = 2, \boldsymbol{\Sigma}_1 = \boldsymbol{\Sigma}_2 = 0.25\mathbb{I}_d, \boldsymbol{p}_1 = -\boldsymbol{p}_2$ and $\boldsymbol{p}_1, \boldsymbol{Q}_\star \sim \mathcal{N}(0, \mathbb{I}_d)$ independently. Here, we use a definite matrix $A$ from (128) , which differs form the one used in the main text. Experiments were conducted as in Fig. 2.

### E.3    More Complex Architectural Choices

As discussed previously in Section C.10, the theory developed in this work can be relaxed to incorporate arbitrary statistical correlations between tokens, changes in the (low-)rank student, generally untied key and query matrices and the presence of a low-rank trainable value matrix. This surplus in complexity comes at the cost of more intricate and heavier equations.

In the present section, we report experiments in the four settings above, but not the theoretical predictions. Fig. 10-13 compare the change in architecture with the setting considered in the main text, for a given $\omega = 0.5$ and varying sample complexity $\alpha$. The exhaustive experiments for varying $\omega$ are shown in Fig. 14 – which immediately shows that the general idea of the phase transition from positional to semantic solution is consistent, but that the shape of the phase transition curve is influenced by the architecture.

For correlated inputs, the transitions seems to moves to larger values of $\alpha$ (Fig. 10). For the case of a student with a rank-2 query matrix, and a teacher with a rank-1 query matrix, we consider three possible initilisations of the columns of $Q$ when finding minima of GD empirically.

Initializing both semantically, both positional, or one of them in each way, as shown in Fig. 11. Interestingly, the dual initialization seems to do approximately as well as the best of the two other initializations. The optimization generally seems more complex, as the phase diagram in the upper right corner of Fig. 14 are not entirely clean, and more noisy than empirical results from other architectures.

When we compare tied and untied $Q$ and $K$ weights in Fig. 12 we observe that the additional parameters we need to learn in the case of an extra $K$ come at a small cost, as the transition to the semantic solution is moved to the right.

Finally, for a value matrix (Fig. 13) the results seem to largely resemble the transition line we observed without the value matrix.

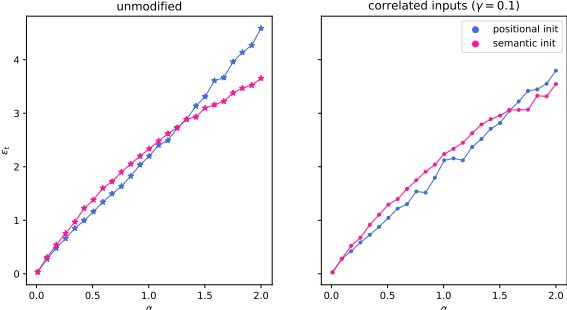

Figure 10: **Uncorrelated vs. correlated inputs** ($\omega = 0.3, \sigma = 0.5, \lambda = 0.001$). In the main text we sample all datapoints $\mathbf{x} \in \mathbb{R}^{d \times L=2}$ such that the columns are independent. We compare this setting with a correlated data structure with a hidden latent: We sample three vectors $u, v, w \sim \mathcal{N}(\mathbf{0}, \mathbf{1})$, and set the first column of $x$ to be $(\sigma u + \gamma v)/\sqrt{\sigma + \gamma}$ and the second one $(\sigma w + \gamma v)/\sqrt{\sigma + \gamma}$. Experiments are repeated 5 times per data point with $d = 1000$.

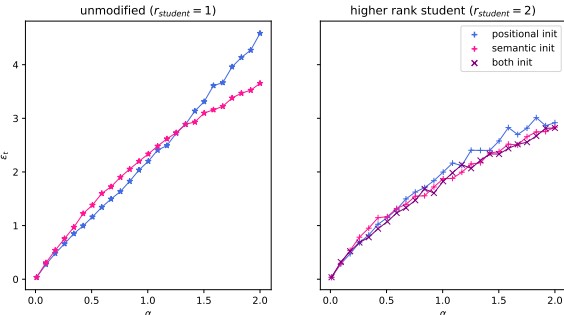

Figure 11: **Rank 1 vs. rank 2 student** ($\omega = 0.3, \sigma = 0.5, \lambda = 0.001$). We compare different initializations of the higher-rank student. Positional is when both columns of the student matrix $\hat{Q}$ are initialized using the positional strategy. We do the same for the semantic strategy. The 'both' initialization initializes one column using the positional strategy and one using the semantic strategy. Experiments are repeated 5 times per data point with $d = 1000$.

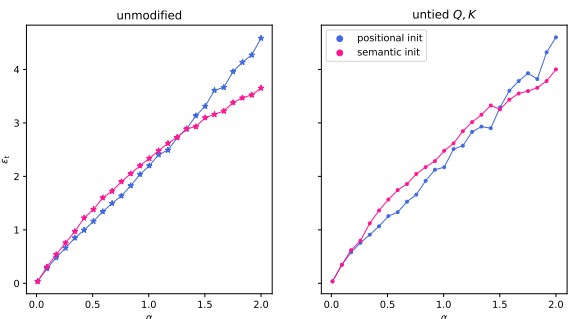

Figure 12: **Tied vs. independent weight Q, K** ($\omega = 0.3, \sigma = 0.5, \lambda = 0.001$). We compare the student setting from the paper, where the query and key matrices are bound to each other with the setting where we set them independently. We initialize with them both being either close to the positional or close to the semantic initialization. The phase transition for the semantic minimum dominating moves to the left, i.e. more samples are now needed. Experiments are repeated 5 times per data point with $d = 1000$.

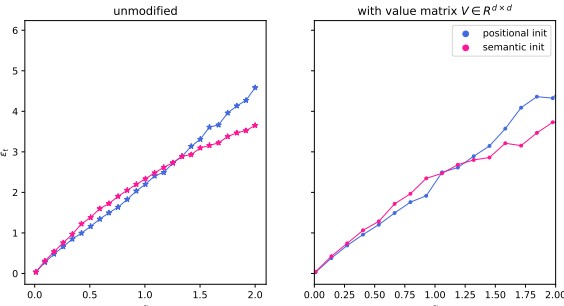

Figure 13: **No value matrix vs. value matrix.** ($\omega = 0.3, \sigma = 0.5, \lambda = 0.001$). We compare the setting from the main text with adding a value matrix, i.e. a trainable parameter $V \in \mathbf{R}^{d \times d}$. This is applied to every embedding before they are averaged over using the attention matrix. We ran the experiment 5 times for each $\alpha$ with $d = 500$, and rescaled the training error to compare to the experiments in the main, where $d = 1000$.

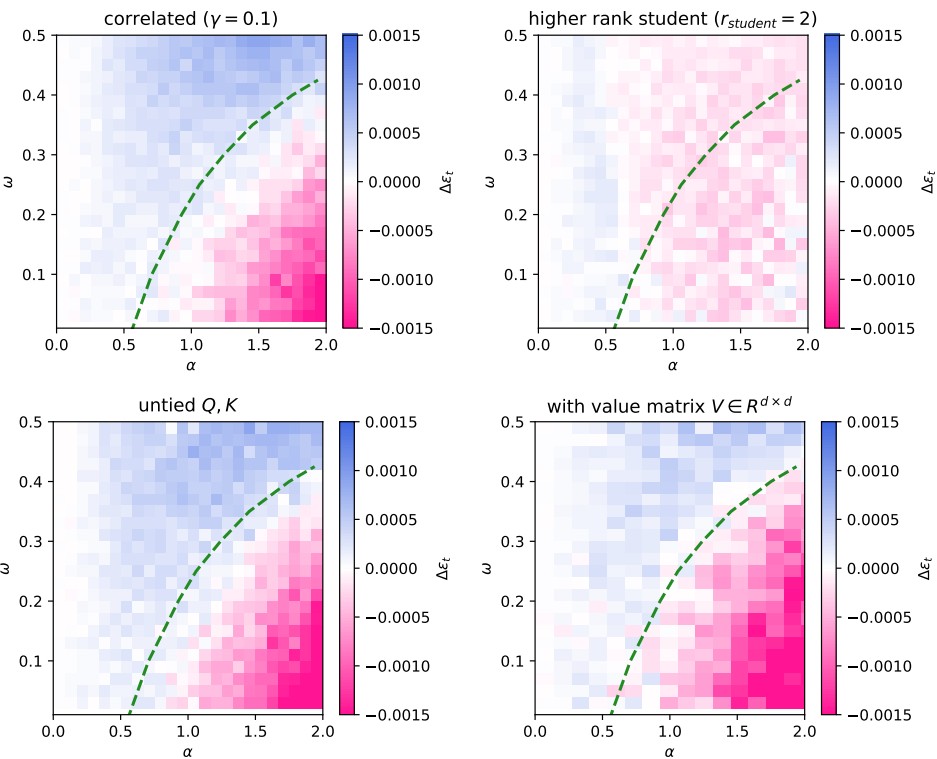

Figure 14: Phase diagrams of the difference in training loss between the semantic and positional solutions from Fig. 10-13, in terms of $\omega$ and the sample complexity $\alpha$. The green dashed line represents the theoretical phase transition which we obtained from the architecture as considered in the main text.

## E.4 Uninformed initialization and training via Adam

In or experiments, to obtain the empirical results, we initialize the GD optimizer in an informed fashion, i.e. initializing $Q_\star$ of the student with $r = 1$ as either $p_1$ (positional) or $Q_\star$ (semantics). GD then converges in the two local optima described by our theory.

Since our theory only ascertains that these solutions predicted are indeed fixed points of GD for large sizes, this does not have direct implications for other types of optimization algorithms. In Fig. 15 we show that indeed running the Adam optimizer from an *uninformed* initialization may lead one to either of the local minima for $d = 100$. For larger $d$ we observe the semantic minimum is reached less often than the positional minimum, and a considerable number of times the algorithm simply does not find either of them.

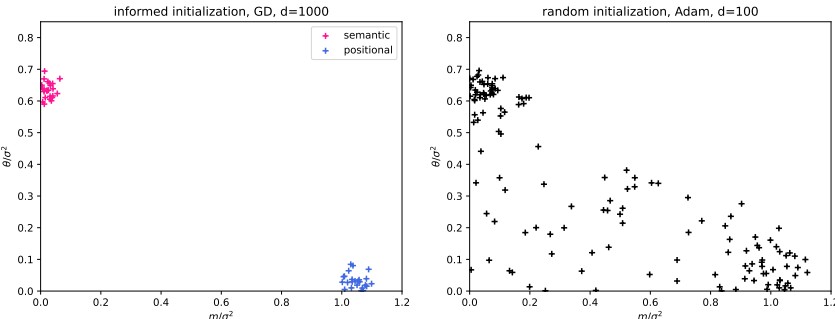

Figure 15: *Comparing GD and Adam.* Settings as in Fig. 2 for the sample complexity $\alpha = 2$. The student parameter **Q** is obtained via either **(left)** positional and semantic informed initialization and **(right)** GD training from a random initialization are compared. Each point represents a single run. For the informed GD, we used the same optimization parameters as in Fig. 2 (24 runs per initialization). For Adam we trained on the same data, but for $2,500$ epochs with learning rate $\eta = 0.01$ (showing 140 runs).

