# OpenReview forum: "A Phase Transition between Positional and Semantic Learning in a Solvable Model of Dot-Product Attention"
_NeurIPS.cc/2024/Conference — NeurIPS 2024 spotlight_

### Official Review · Reviewer_wGcv · 2024-07-12

**Soundness:** 3
**Presentation:** 2
**Contribution:** 3
**Rating:** 6
**Confidence:** 2

**Summary:**

The authors consider a simplified attention network with shared Query and Key matrices trained with an MSE loss and show a sharp phase transition exists when training this network. In the high-dimensional limit, this paper provides a closed form solution to the training and test loss and shows that a phase transition exists in terms of sample complexity where the model goes from solving the solution with positional information (locations of the tokens in the sentence) to semantic information (content of the tokens in the sentence). The authors show that this theoretical result shows the advantage of the attention mechanism over a fully-connected network for this task with sufficient data.

**Strengths:**

Understanding the properties that lead to phase transitions in neural networks, and more broadly understanding training dynamics in transformer models is an important area of research. This work provides the first theoretical result showing phase transitions existing in attention mechanisms from learning, and thus opens the door for more work on the learning dynamics of transformers. I think the findings of this paper are important for interpretability research, but since it is not my area, I'm not able to strongly recommend it one way or the other.

The design for the task and properties resulting in the phase transition are clear and simple. The authors are able to empirically test their results and find that models training on data nearly match the theory. The purely positional baseline provides a nice comparison for the dot product attention mechanism which *can* modulate between positional and semantic information as a result of the phase transition. Other empirical results are well justified and presented clearly.

**Weaknesses:**

Because the results rely on the model reaching the minimum, it is unclear how well these results extend to randomly initialized networks. This is more a problem for studying training dynamics, though, which this paper does not aim to address.

some parts of the paper are not well motivated or presented. In particular 4.1 and 4.2 may only reach a small audience without more description.

The authors make many simplifications on the attention mechanism

**Questions:**

Do the authors have speculations about how changes to the current architecture would affect the phase transition? For example, if the value matrix was not the identity, would the model be likely to transition to the semantic solution faster?

**Limitations:**

Yes

---

> ### Author Rebuttal · Authors · 2024-08-06
>
> We thank the reviewer for their careful reading of our work and insightful comments, which we address below.
>
> > Because the results rely on the model reaching the minimum, it is unclear how well these results extend to randomly initialized networks. This is more a problem for studying training dynamics, though, which this paper does not aim to address.
>
> We completely agree that further understanding which of the characterized minima is reached by a given optimizer under given conditions (in other words, the dynamics) is of great interest. As evidenced in [9] in a related setting, the answer to this question can depend on many optimization hyperparameters. While we give some elements of answer in Appendix D.3, we believe a thorough answer to this question warrants a careful empirical investigation, and theoretical analysis, of the training dynamics. This significant research endeavour however falls out of the scope of this first work-- whose primary focus is indeed to provide the first theoretical analysis of a phase transition in the learning of attention models, as appreciated by all three reviewers. We will however further emphasize the importance of this future direction in the conclusion of the revised manuscript.
>
> > some parts of the paper are not well motivated or presented. In particular 4.1 and 4.2 may only reach a small audience without more description.
>
> We will take advantage of the allowed extra page in the camera-ready version to include further intuition-building discussion below 4.1 and 4.2, to describe in more detail the content and consequences of these technical statements.
>
> > The authors make many simplifications on the attention mechanism
>
> The results are indeed stated under four main simplifying assumptions, namely (a) uncorrelated tokens, (b) tied key and query weights (c)  value weights set to identity and no readout (d) all weights are low-rank . Assumptions (a,b,c) can in fact be relaxed, and the analysis can be extended to incorporate arbitrary statistical correlations between tokens, generically untied key and query weights, and accomodate a low-rank trainable value matrix -- at the price of much more intricate and heavier equations (see also our answers to Reviewers uCdk and QM4T). For this reason, we have chosen for the sake of clarity to restrict the discussion to a simpler case, already exhibiting the rich phenomenology of semantic versus positional learning. We will highlight these generalizations in Appendix A of the revised manuscript, and include a pointer thereto in the main text.
>
> Assumption (d) is on the other hand required for the analysis. Note however that weights are enforced to be low-rank in a number of practical settings in language modeling, notably in the context of model compression [a] or finetuning [b], to ease the costs induced by finetuning or deploying large language models, see also our answer to reviewer QM4T.
>
> [a] Hsu et al, language model compression with weighted low-rank factorization, ICLR 2022.
>
> [b] Hu et al, LoRA: Low-Rank Adaptation of Large Language Models, ICLR 2022.
>
> > Do the authors have speculations about how changes to the current architecture would affect the phase transition? For example, if the value matrix was not the identity, would the model be likely to transition to the semantic solution faster?
>
> After the reviewer's question, we ran some preliminary experiments that suggest that including a learnable value matrix leads to no sensible change to the phase transition. These experiments are illustrated in Fig. 4 of the attached pdf, where we also include experiments illustrating the effect of other architectural changes (see also the answers to reviewers uCdk and QM4T). We will include all these figures, and a discussion thereof, in the final version of the manuscript.

---

> > ### Comment · Reviewer_wGcv · 2024-08-12
> > **Thank you for the reply**
> >
> > Thank you for the detailed reply. I appreciate the pointer to appendix D.3 which is helpful. Besides that, I definitely agree this would be out of the scope of the current work.
> >
> > P2: Thank you the extra explanation will be helpful. Again, I am sympathetic that it simply isn't possible to catch everyone up in such a short space, I have found that adding the extra explanations has been worth it, though.
> >
> > I think the authors properly address the concerns, and I'm a bit more confident after the followup discussion and reading the other reviews

---

### Official Review · Reviewer_QM4T · 2024-07-12

**Soundness:** 4
**Presentation:** 3
**Contribution:** 3
**Rating:** 7
**Confidence:** 2

**Summary:**

This paper introduces a simplifed model of attention and analyzes it theoretically, showing that there exists a phase transition between a paradigm where attention is based mostly on position to one where it is not (which they call "semantic").  I will confess to not being an expert on the methods used and so did not follow the main results (which take about a page just to state) and proofs in detail.  They strike me, however, as genuinely useful and insightful, albeit with a caveat or two about some of the assumptions needed to get them to work (e.g. sequential independence).

**Strengths:**

* Provides theoretical analyses of a model of self-attention, finding a closed-form solution.
* Asymptotic analysis demonstrates a phase-shift between two minima, one which relies on position and one which does not.
* The first analysis of this type to an attention layer, instead of just a feed-forward layer.

**Weaknesses:**

* Very dense mathematically, so hard to follow for a reader not intimately familiar with the literature to which it contributes.
* While toy models are indeed useful objects to study in general, there are unclear connections between some of the assumptions (e.g. independent samples of individual tokens, and the low-rank attention) and actual language modeling practice.

**Questions:**

* How much do you think the results depend on the nature of the data? I'm thinking in particular of the fact that words are drawn _independently_: could this be part of why positional information becomes irrelevant, since the distribution at each position is the same?  Do you have any expectations for slightly more realistic settings (even, e.g. independent $n$-grams instead of unigrams)?

**Limitations:**

Yes

---

> ### Author Rebuttal · Authors · 2024-08-06
>
> We thank the reviewer for their reading of our work, and the many interesting questions, which we answer below.
>
> > Very dense mathematically, so hard to follow for a reader not intimately familiar with the literature to which it contributes.
>
> We will take advantage of the additional page allowed in the camera-ready version to provide further intuition-building discussion below the statement of Result 4.2, alongside further context and discussion on how our contribution fits in the broader literature in the related works section.
>
> > While toy models are indeed useful objects to study in general, there are unclear connections between some of the assumptions (e.g. independent samples of individual tokens, and the low-rank attention) and actual language modeling practice.
>
> We thank the reviewer for raising this important point. The assumption of independent tokens was actually made for the sake of clarity and conciseness of presentation, and can be relaxed in the analysis to include generic statistical correlations between the tokens, as we further detail in our answer to the following question.
>
> Low-rank attention weights have been considered in practice in language modeling in the context of model compression, see for example [a]. In this approach, the weights of large language models are approximated by low-rank matrices to reach a smaller model, easier to fine-tune and deploy. The idea to train low-rank weights, at least at the finetuning stage, also underlies the celebrated LoRA scheme [b], which allows for the resource-efficient yet effective fine-tuning of large language models.
>
> [a] Hsu et al, language model compression with weighted low-rank factorization, ICLR 2022.
>
> [b] Hu et al, LoRA: Low-Rank Adaptation of Large Language Models, ICLR 2022.
>
> > How much do you think the results depend on the nature of the data? I'm thinking in particular of the fact that words are drawn independently: could this be part of why positional information becomes irrelevant, since the distribution at each position is the same? Do you have any expectations for slightly more realistic settings (even, e.g. independent
> n-grams instead of unigrams)?
>
> The reviewer's intuition is right that the amount of correlation between tokens affects the phase transition. In Fig.1 of the attached pdf (see global rebuttal), we provide additional numerical experiments showing how introducing more correlation between tokens shifts the phase transitions to higher sample complexities, i.e. increases $\alpha_c$. On an intuitive level, this is because each data point carries less semantic information (as correlations make tokens more redundant), and therefore more data points are needed to identify the semantic content, and learn a semantic mechanism. By the same token, we generically expect that the phase transition happens at higher sample complexities for n-grams than for unigrams.
>
> We would also like stress that the analysis can, in fact, be extended to cover arbitrary statistical correlations between different tokens, albeit at the price or heavier equations (see also the answer to Reviewer uCdk). For these reasons, we have chosen for the sake of clarity not to discuss the effect of these correlations, and focus on the uncorrelated case, which already yields a very rich phenomenology in terms of semantic versus positional learning. We will however include a discussion of this generalization in Appendix A of the camera-ready version, alongside the aforedescribed figure.

---

> > ### Comment · Reviewer_QM4T · 2024-08-12
> >
> > Thanks!  I really appreciate these clarifications and am looking forward to reading the generalization Appendix in a camera-ready version.

---

### Official Review · Reviewer_uCdk · 2024-07-13

**Soundness:** 3
**Presentation:** 3
**Contribution:** 3
**Rating:** 7
**Confidence:** 3

**Summary:**

The authors state an asymptotic result characterizing the test MSE and training loss in a simplified single-layer model of dot product attention. They apply this result to study a special case in which the target attention function contains a tradeoff parameterized by $\omega$ between positional (i.e. dependent only on index location) and semantic (i.e. input-dependent) terms. Analysis of the solution characterized in the theoretical result shows that for a fixed $\omega$, there is a sharp boundary in terms of the sample complexity $\alpha$ (ratio of sample size to embedding dimension) between a semantic vs. positional parameter as the global minimum. These results are corroborated by an empirical analysis illustrating the distinct minima and comparing the empirical difference in train loss at semantic vs positional minima to the theoretical prediction as a function of $\alpha$ and $\omega$.

**Strengths:**

The paper contributes an original result on the learning theory of attention models, along with a creative and insightful application of this result to a setting that contrasts positional versus semantic solutions. The result appears significant as a theoretical lens through which to characterize the loss landscape of an attention model, and it may have applications beyond the specific positional-vs-semantic target model studied in this paper. The exposition and empirical illustration of the result are clear.

**Weaknesses:**

As noted by the authors in the limitations, the stated result applies to a simplified model both in terms of the structure of the attention function and in terms of the input data.

It is unclear how the solutions of (7) were found in practice, as discussed in Section 5, or how it was determined that the global minimum was among this pair of fixed points for the values of $(\alpha, \omega)$ studied in the experiments.

**Questions:**

Can the authors comment on why the simplifications in the attention model (value weights set to identity, key and query weights tied) are required for their result?

The positional-to-semantic phase transition detailed in Section 5 is discovered as a consequence of Theorem 4.2 applied to a specific model (Eq (14)). Have the authors considered other models or aspects of attention-based learning that could be studied through the same lens?

**Limitations:**

Limitations are identified and discussed in the paper.

---

> ### Author Rebuttal · Authors · 2024-08-06
>
> We thank the reviewer for their appreciation of our work and their constructive questions, which we address below.
>
> > the stated result applies to a simplified model [...] in terms of [...] the input data.
>
> We have indeed chosen for the sake of clarity to present the results for the simplest instance of input data distribution --namely uncorrelated Gaussian tokens--, for which the trained attention can learn both positional and semantic solutions. The analysis can however be extended to include statistical correlations between the tokens, and cover more structured Gaussian mixture token distributions, see also the answer to reviewer QM4T, at the price of much heavier equations. For this reason, we chose not to discuss this general case for the sake of clarity. We generically expect that introducing correlations leaves the phenomenology qualitatively unchanged, but shifts the phase transition towards higher sample complexities (i.e. increases $\alpha_c$), as we illustrate in Fig. 1 of the attached pdf. We will include these additional theoretical discussions, alongside the additional figure, in the final revision of the manuscript.
>
> > Can the authors comment on why the simplifications in the attention model (value weights set to identity, key and query weights tied) are required for their result?
>
> The analysis can actually be extended to include trainable value weights, as well as untied key and query weights, provided they remain low-rank. On the other hand, these extensions come at the price of more cumbersome equations -- for instance, the size of the summary statistics matrices of Result 4.2 triple, and the expression of the Moreau envelope (l.168) becomes sizeably more intricate. For these reasons, we chose for the sake of clarity and conciseness to present our result under these various simplifications. We shall include a detailed discussion on how the analysis can be generalized to these cases in the camera-ready version of the manuscript.
>
> We illustrate in attached pdf how these architectural changes affect the phase transition : Fig. 3 shows how untying the key and query weights keeps the phase transition but shifts it to towards higher sample complexities (i.e. larger $\alpha_c$). Fig. 4 shows how appending a trainable value matrix leads to no sensible change to the phase transition.
>
> > It is unclear how the solutions of (7) were found in practice, as discussed in Section 5, or how it was determined that the global minimum was among this pair of fixed points for the values of $(\alpha, \omega)$
>  studied in the experiments.
>
> We will include further discussion on these two points in the final revision of the manuscript. The solutions of (7) were found by numerically iterating the set of self-consistent equations (7) until convergence, for various initializations. The training loss of the different fixed points thus reached was then evaluated using equation (12) of Result 4.2. We found in all examined settings, both in the theory and in experiments, that the lowest training loss was achieved by one of the two fixed points (positional or semantic) discussed in Section 5, and thus concluded that the global minimizer belongs to this pair of fixed points.
>
> > The positional-to-semantic phase transition detailed in Section 5 is discovered as a consequence of Theorem 4.2 applied to a specific model (Eq (14)). Have the authors considered other models or aspects of attention-based learning that could be studied through the same lens?
>
> As the reviewer correctly surmises, Result 4.2 provides a very versatile playground to explore other models of attention-based learning, beyond the model considered in the present manuscript. Other aspects of attention mechanism which can be directly explored through the lens of Result 4.2. include the use of causal masks, and cross-attention mechanisms. A thorough study theoreof however warrants separate works, and fall out of the scope of the current manuscript, whose focus is on the interplay between positional and semantic learning. We will however include a highlight of these other aspects in the conclusion section of the revised manuscript.

---

> > ### Comment · Reviewer_uCdk · 2024-08-13
> >
> > Thanks to the authors for their comments. These additional details address the main points of my review and the corresponding (minor) updates to the manuscript will further help contextualize the main result. I am happy to support this paper for acceptance.

---

### Author Rebuttal · Authors · 2024-08-06

Thank you all for taking the time to read and review our work.

In this global response we post the pdf file containing plots of some preliminary experiments that help to clarify several questions raised in the reviews.
We refer to this global file in the separate responses to each reviewer.

Precisely, we show how the phase transition between positional and semantic minima moves empirically for:
1. Uncorrelated vs. correlated inputs,
2. Rank 1 vs. rank 2 student,
3. Tied vs. independent weight Q, K,
4. No value matrix vs. value matrix.

We will provide more extensive versions of the same experiments in the appendix of a camera-ready version.

---

### Decision · Program_Chairs · 2024-09-25

**Decision:**

Accept (spotlight)

**Comment:**

The paper presents an asymptotic result on a simplified model of attention, and use it to show that, given certain assumptions, a sharp transition between a more order-dependent and a more "semantic" type of attention mechanism will emerge.

As with all theoretical papers based on simplified models, it is not entirely clear the extent to which the presented insights generalize to real-life models, but all reviewers agree that this is an exciting paper worth being presented at the conference.